# Global emergence of a hypervirulent carbapenem-resistant *Escherichia coli* ST410 clone

Xiaoliang Ba [1,8], Yingyi Guo [2,8], Robert A. Moran[3], Emma L. Doughty[3], Baomo Liu[4], Likang Yao[2], Jiahui Li[2], Nanhao He[2], Siquan Shen[5,6], Yang Li[7], Willem van Schaik [3], Alan McNally [3], Mark A. Holmes [1,9] ✉ & Chao Zhuo [2,9] ✉

Carbapenem-resistant *Escherichia coli* (CREC) ST410 has recently emerged as a major global health problem. Here, we report a shift in CREC prevalence in Chinese hospitals between 2017 and 2021 with ST410 becoming the most commonly isolated sequence type. Genomic analysis identifies a hypervirulent CREC ST410 clone, B5/H24RxC, which caused two separate outbreaks in a children's hospital. It may have emerged from the previously characterised B4/H24RxC in 2006 and has been isolated in ten other countries from 2015 to 2021. Compared with B4/H24RxC, B5/H24RxC lacks the $bla_{OXA-181}$-bearing X3 plasmid, but carries a F-type plasmid containing $bla_{NDM-5}$. Most of B5/H24RxC also carry a high pathogenicity island and a novel O-antigen gene cluster. We find that B5/H24RxC grew faster in vitro and is more virulent in vivo. The identification of this newly emerged but already globally disseminated hypervirulent CREC clone, highlights the ongoing evolution of ST410 towards increased resistance and virulence.

Antimicrobial resistance (AMR) has become one of the most serious public health problems of the 21st century, threatening the effective prevention and treatment of infections in humans and animals alike[1,2]. A bacterial strain is considered multidrug-resistant (MDR) if it is resistant to at least one agent in three or more antimicrobial classes[3]. Carbapenems are a group of last-line antibiotics for treating infections caused by MDR Gram-negative bacteria[4]. However, carbapenem resistance has been observed in many pathogens, including members of the Enterobacteriaceae family such as *Escherichia coli*[4,5]. The most frequently observed carbapenem resistance mechanism in Enterobacteriaceae is the production of carbapenemases that hydrolyse

carbapenems[4,6]. *Klebsiella pneumoniae* carbapenemase, New Delhi metallo-β-lactamase (NDM), imipenemase, Verona integron-encoded metallo-β-lactamase, and oxacillinase-48 (OXA-48) are the most commonly encountered carbapenemases in Enterobacteriaceae[7–9]. Among these, NDM has been reported to be highly prevalent in many parts of the world[9–11].

In the past two decades, carbapenem-resistant *E. coli* (CREC) has emerged rapidly and become increasingly prevalent around the globe, causing various hard-to-treat clinical infections[9,10,12]. The global CREC population is made up of diverse multilocus sequence types (STs) with varied geographic distributions. A previous study reported two major

[1]Department of Veterinary Medicine, University of Cambridge, Cambridge, United Kingdom. [2]State Key Laboratory of Respiratory Disease, First Affiliated Hospital of Guangzhou Medical University, Guangzhou, China. [3]Institute of Microbiology and Infection, College of Medical and Dental Sciences, University of Birmingham, Birmingham B15 2TT, United Kingdom. [4]Department of Pulmonary and Critical Care Medicine, The First Affiliated Hospital of Sun Yat-sen University, Guangzhou, China. [5]Institute of Antibiotics, Huashan Hospital, Fudan University, Shanghai, China. [6]Key Laboratory of Clinical Pharmacology of Antibiotics, Ministry of Health, Shanghai, China. [7]Department of Clinical Laboratory, Children's Hospital of Soochow University, Suzhou, Jiangsu, China. [8]These authors contributed equally: Xiaoliang Ba, Yingyi Guo. [9]These authors jointly supervised this work: Mark A. Holmes, Chao Zhuo. ✉e-mail: mah1@cam.ac.uk; chaosheep@sina.com

(>10%) STs (ST410 and ST131) and three minor (5–10%) STs (ST1248, ST167, and ST405) in a global collection of 229 CREC collected by two global surveillance programs during 2015–2017[13]. No CREC from China was reported in that study[13], although CREC isolates are frequently isolated in Chinese healthcare settings. Two Chinese national surveillance studies carried out over a similar period (2014–2017) revealed that most CREC isolates in China belonged to ST167 or ST131, with ST410 the third most prevalent[9,14].

*E. coli* ST410 is an extraintestinal pathogen associated with multidrug resistance, and has been recognised as a high-risk international clone[15,16]. Whole-genome sequence analysis and evolutionary reconstruction of *E. coli* ST410 revealed that ST410 is comprised of two lineages, namely lineage A with *fimH*53 (A/H53) and lineage B with *fimH*24 (B/H24)[15]. The B/H24 lineage is further divided into four sub-lineages: B1/H24, B2/H24R, B3/H24Rx and B4/H24RxC. Sub-lineage B2/H24R is characterised by fluoroquinolone resistance associated with mutations in *gyrA* and *parC*, B3/H24Rx is defined by the introduction of the extended-spectrum β-lactamase-encoding gene $bla_{CTX-M-15}$, and B4/H24RxC is defined by a further introduction of the carbapenem resistance gene $bla_{OXA-181}$ carried on an IncX3 plasmid[15]. A more recent study has proposed a modification to this classification, where ST410-B1 is identical to B1/H24, ST410-B2 includes both B2/H24R and B3/H24Rx, and ST410-B3 is identical to B4/H24RxC[17]. That study characterised ST410-B2 by the introduction of fluoroquinolone resistance mutations and ST410-B3 by the acquisition of a four-amino-acid (YRIN N337N) insertion in the *ftsI*-encoded penicillin-binding protein 3 (PBP3, also called FtsI)[17]. The YRIN insertion in PBP3, together with YRIK and TIPY insertions, were reported to confer reduced susceptibility to a broad range of β-lactams such as ceftazidime, cefepime and aztreonam[18,19]. Chen et al. reported that nearly all (292/293) ST410-B3 isolates analysed contain the *ftsI* YRIN insetion[17]. Over the past decade, there have been increasing reports of serious infections and possible hospital outbreaks involving the carbapenem-resistant ST410 sub-lineage B4/H24RxC in both developed and low- and middle-income countries[15,16,20–23].

In this work, we investigate the population of clinical CREC isolates from Chinese hospital patients between 2017 and 2021. The most commonly isolated CREC lineage, ST410, is analysed further. We examine the global ST410 population structure by comparing 847 publicly available ST410 genomes to the 109 ST410 genomes generated in this study. A hypervirulent MDR ST410 clone is identified and designated B5/H24RxC. Comparative genomic analyses and phenotypic assays are performed to characterise B5/H24RxC and provide insights into its emergence and evolution.

## Results

### Characteristics of CREC isolates from Chinese hospitals

A total of 388 CREC isolates were collected from hospitals across 26 Chinese provinces between 2017 and 2021 (Fig. 1a, c; Supplementary Data 1). The isolates were recovered from various clinical samples, including ascites, bile, blood, bronchoalveolar fluid, pus, wound secretion, sputum and urine. The most common clinical sample types were urine ($n = 111$), sputum ($n = 64$) and blood ($n = 47$) (Fig. 1b), indicating possible associations between CREC and urinary tract infections (UTIs), pneumonia and bloodstream infections. All CREC isolates were resistant to at least one of the carbapenem antibiotics tested. The isolates had a median MIC for imipenem of 8 mg/L (range 0.125 to >128 mg/L), a median MIC for meropenem of 32 mg/L (<0.03 to >128 mg/L) and a median MIC for ertapenem of 32 mg/L (<0.03 to >128 mg/L) (Fig. 1f; Supplementary Data 1). It is to be noted here that analysis for ertapenem MIC was based only on the 168 isolates collected by Guangzhou Medical University, excluding isolates collected by China Antimicrobial Surveillance Network (CHINET) as ertapenem MIC was not routinely determined by CHINET. Amongst the isolates tested for tigecycline and polymyxin resistance, 40 were resistant to

tigecycline (MIC, 1–4 mg/L) and 12 were resistant to polymyxin B (MIC, 4–64 mg/L) (Supplementary Data 1).

Genomic analysis revealed that the most prevalent carbapenem resistance gene in this CREC collection was $bla_{NDM-5}$ ($n = 282$), followed by $bla_{NDM-1}$ ($n = 42$) (Fig. 1e; Supplementary Data 1). Interestingly, only 87.1% (338/388) of the CREC isolates possessed at least one carbapenem resistance gene, suggesting alternative carbapenem resistance mechanisms in the rest of the isolates. Eleven isolates carried the colistin resistance gene *mcr-1* (Supplementary Data 1). The CREC isolates belonged to 71 STs with 16 of them not assigned an ST. The most commonly isolated STs were ST410 ($n = 109$), ST167 ($n = 41$), ST131 ($n = 12$) and ST617 ($n = 12$) (Fig. 1d). This indicated a change in CREC population in China relative to studies conducted between 2015 and 2017[9,14], where ST410 was reported to be the third most commonly isolated CREC ST behind ST131 and ST167.

### Outbreaks of a CREC ST410 clone in a children's hospital

Of the 109 ST410 CREC, 49 were isolated from 47 inpatients admitted to a children's hospital in eastern China between 2018 and 2020. Most of the patients ($n = 33$, 70.2%) were under 60 days of age and 72.3% ($n = 34$) of them were female (Fig. 2b). The patients were admitted to different hospital departments and suffered from various infections, including UTI ($n = 31$, 66.0%), pneumonia, septicaemia and bacteraemia (Supplementary Data 3). Patients were grouped into two cohorts according to their time of admission and the length of hospital stay (Fig. 2a). One cohort stayed in the hospital from March 2018 to August 2019 and the other from January 2020 to September 2020.

Four groups of *E. coli* ST410 were identified in this children's hospital, characterised by SNP analysis relative to the complete genome of isolate 19-7 (Genbank: CP123017 to CP123023) which was from a patient with UTI in the same hospital (Fig. 2c). A pairwise comparison of the SNP differences across all isolates showed that isolates within the two groups differed by no more than 23 SNPs, suggesting possible outbreaks in the hospital based on a previously recommended threshold of ≤25 SNPs[24]. The outbreaks happened within multiple hospital departments as well as inter-departmentally as indicated by where and when the patients stayed during their treatment (Fig. 2a). Group-1 included 20 isolates mainly from between March 2018 and August 2019, with the exception of isolate 20-20 from June 2020, while group-2 only included 27 isolates from 2020 (Fig. 2a, c). The pairwise comparison also revealed that there were just 29–69 SNPs between isolates in these two groups. Singletons 18-4 and 18-10 did not belong to either group and each formed a group of their own, but 18-4 was closely related with the isolates in the two main groups as indicated by a pairwise SNP distance of 29 to 48. However, isolate 18-10 was distant from all other isolates with a pairwise SNP distance of 215–240. The dividing of the groups was further supported by a Fastbaps analysis of the phylogeny of the isolates, although 18-4 was separated from Group-1 at the second Fastbaps level (Fig. S3c, S3d). Apart from isolate 18-10, the CREC ST410 in this children's hospital were found to be distinct from previously reported ST410 clones and therefore were analysed further to define their genomic characteristics and determine their origins.

### Global population structure of *E. coli* ST410 reveals an MDR clone, B5/H24RxC

In order to analyse the CREC ST410 isolated from the children's hospital in a global context, we constructed a phylogeny of 956 *E. coli* ST410 isolates based on a core-genome SNP alignment (Fig. 3a). Fastbaps clustered this global collection into 11 BAP groups based on the SNP alignment and phylogeny. Most of the isolates were assigned to two of the main groups, BAP1 ($n = 429$) and BAP2 ($n = 434$), which made up 90.3% of the collection. The previously reported MDR clone B4/H24RxC, which carries the $bla_{OXA-181}$ carbapenemase gene in an X3 plasmid, was found to belong to BAP1. All of the CREC ST410 isolates

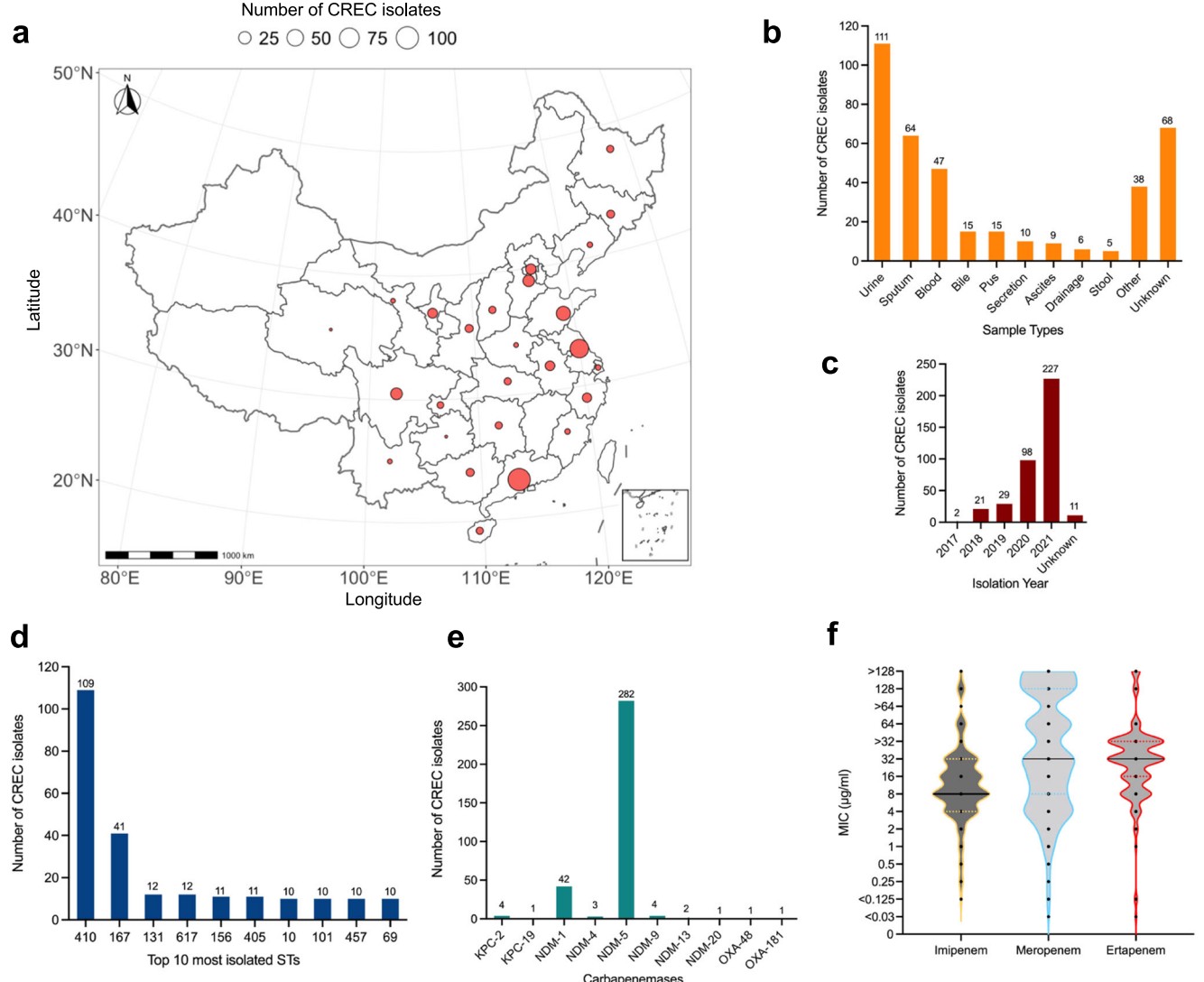

**Fig. 1 | Characteristics of CREC isolates ($n = 388$) from Chinese hospitals. a** The geographical distribution of the CREC isolates collected in this study shown on the map of P. R. China. **b** Bar chart showing the various types of clinical samples used in this study for the isolation of CREC. **c** Bar chart showing the distribution of isolation years. **d** Bar chart showing the 10 most identified STs in this CREC collection. **e** Bar chart showing identified carbapenem-resistant genes in this CREC collection. **f** Violin plot showing the distribution of MICs of imipenem, meropenem and ertapenem for the CREC isolates, the median MIC for each carbapenem antibiotics is represented with a black line. The plot for ertapenem MIC was generated based on data for isolates ($n = 168$) in the Guangzhou Medical University (GMU) collection. Source data are provided as a Source Data file.

from the children's hospital outbreaks, apart from isolate 18-10, also belonged to group BAP1. However, the children's hospital isolates clustered into a subclade that was distinct from the B4/H24RxC clone. This subclade, here designated B5/H24RxC ($n = 174$), rarely contained the X3 plasmid-associated $bla_{OXA-181}$ of B4/H24RxC. Instead, 97.7% (170/174) of the B5/H24RxC genomes contained a F-type plasmid (Fig. 3a, S2), carrying the $bla_{NDM-5}$ carbapenemase gene. Another obvious difference between the two clones was that most B5/H24RxC genomes lacked a small plasmid with a pColKP3-like replicon, which was present in the majority of B4/H24RxC genomes (Fig. 3a, S2).

Further comparative analyses of the two clones showed that B5/H24RxC contained a larger (two-tailed unpaired $t$ test, $p = 0.047$) number of acquired antimicrobial resistance genes (ARGs) and mutations that confer resistance (Fig. 3b; Supplementary Data 4). B5/H24RxC isolates were found to have more (two-tailed unpaired $t$ test, $p < 0.0001$) putative virulence genes than B4/H24RxC isolates (Fig. 3c). B5/H24RxC isolates had a median of 176 putative virulence genes (range 159–190) while the B4/H24RxC clone had a median of 166 (143–181). The difference was mainly caused by the presence of the

high pathogenicity island (HPI) that was originally identified in *Yersinia enterocolitica*[25,26], in 77.6% (135/174) of the B5/H24RxC clone but in none of the isolates of the B4/H24RxC clone (Fig. S2). The HPI in these isolates contained all 11 virulence-associated genes (*fyuA, irp1, irp2, ybtA, ybtE, ybtP, ybtQ, ybtS, ybtT, ybtU,* and *ybtX*). In silico serotyping revealed that, 99.0% (384/388) of all isolates of both clones produced a H9 flagellar antigen, but the majority (95.3%, 204/214) of the B4/H24RxC isolates had an O8 lipopolysaccharide while the majority (74.7%, 130/174) of B5/H24RxC isolates had an Onovel1 (as labelled in the EcOH database by Ingle et al.[27]) lipopolysaccharide (Fig. 3d). BLAST analysis revealed that Onovel1 matched (96.3% coverage and 98.66% identity) a previously reported O-antigen OgN5 (LC177549.1) identified in enterotoxigenic *E. coli* of various STs[28] (Fig. 3e).

The isolates of the B5/H24RxC clone in this international collection were isolated between 2015 and 2021 from humans, food and companion animals across 11 countries on 5 continents (Fig. S1; Supplementary Data 2). The majority of the isolates (89.1%, 155/174) were from countries in eastern and south-eastern Asia, such as China ($n = 78$) and Thailand ($n = 71$).

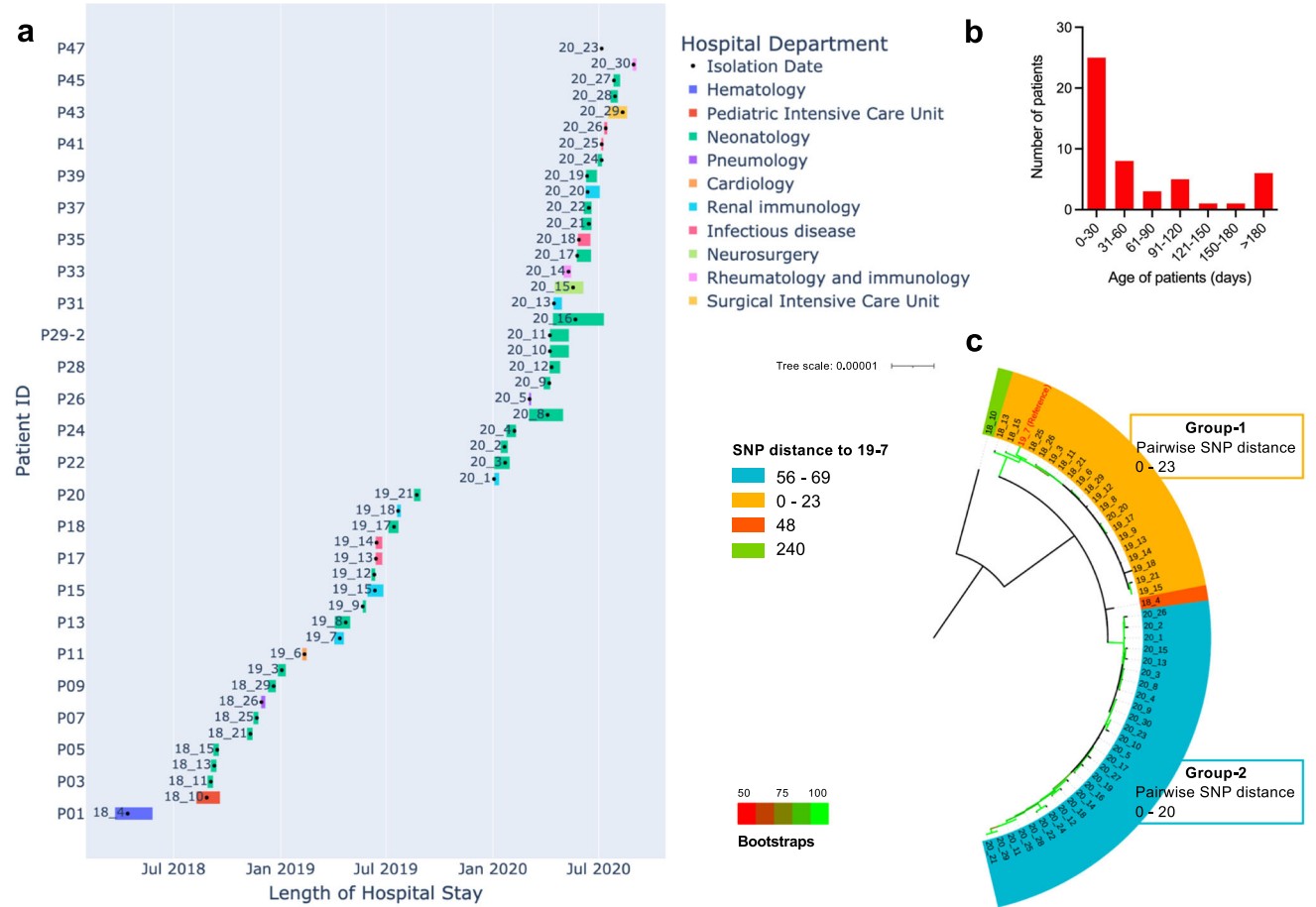

**Fig. 2 | Outbreaks of a ST410 lineage in a children's hospital. a** Gantt plot showing the length of hospital stay of the patients in the children's hospital in eastern China. Patient ID are presented on the *y* axis and the length of stay of each patient is represented with coloured bars. A black dot within the coloured bars indicates the time of the isolation of the isolates. **b** Bar chart showing the age distribution of the patients. **c** Maximum-likelihood core-genome SNP phylogeny of the 49 ST410 CREC isolates in the children's hospital. Colours indicates the SNP distance to the reference genome 19-7. Bootstrap values are represented by gradient colours. Source data are provided as a Source Data file.

## Emergence of the B5/H24RxC MDR clone driven by recombination and horizontal gene transfer

A smaller scale phylogeny reconstruction was performed only on the B4/H24RxC and the B5/H24RxC MDR clones using the complete genome of a previously reported isolate 020026 (B4/H24RxC; Genbank: CP034954 to CP034958) as the reference. Gubbins identified several recombination regions associated with the B5/H24RxC MDR clone, including a 5.7-kb region that contained genes of unknown function in all 174 isolates of the clone (Fig. S5). By excluding SNPs within the recombination regions, the average pairwise distance among the 174 isolates of B5/H24RxC MDR clone was 41, ranging from 0 to 136 (Supplementary Data 5), suggesting likely ongoing global dissemination of this recently emerged clone. A total of 204 SNPs were identified on the branch separating the B5/H24RxC MDR clone from its most closely related isolate (Fig. S5), of which 171 were within the recombination regions, giving a per site r/m ratio (the probabilities that a given site was altered through recombination and mutation) of 5.18. This suggests that homologous recombination contributed to the evolutionary events resulting the emergence of the B5/H24RxC MDR clone. Further comparative analysis of the recombination regions in both clones also revealed the O-antigen gene cluster (O-AGC) switch from O8 in B4/H24RxC to Onovel1 in B5/H24RxC, and the HPI gene cluster in B5/H24RxC clone (Fig. 3e).

Both clones possessed a F-type plasmid containing FII-1, FIA-1, and FIB-49 replicons. The plasmid consisted of a backbone and an antibiotic resistance region. The resistance region was bounded at one end by a copy of IS*1* and at the other end by a partial copy of Tn*5403* (Fig. 4a). Most differences between variants of this F-type plasmid were within the resistance region, while their backbones were almost identical (Fig. 4a). The conserved backbone contained three replicons and genes associated with stability (restriction-modification and toxin-antitoxin systems). Although it contained *finO* and *traX* genes, these were the only remnants of a F-like transfer region and this plasmid was not expected to be conjugative. Outside the resistance region, the backbone was only interrupted by two insertion sequences, IS*Ec12* and IS*1* (Fig. 4a). IS*Ec12* was flanked by a target site duplication but IS*1* was not, suggesting that it had mediated a deletion event post-insertion. Importantly, 695 bp immediately adjacent to the left end of the resistance region were absent from plasmid variants found in B5/H24RxC (Fig. 4a), and appeared to have been lost in a deletion event mediated by the IS*1* at the boundary of the resistance region. This deletion was clear evidence that the backbone variant present in B4/H24RxC was ancestral to that in B5/H24RxC. The resistance region was comprised of a series of ARG-containing translocatable element sequences interspersed with copies of IS*26* (Fig. 4b). Resistance region variants present in the B5/H24RxC clone included an IS*26*-flanked segment that contained *bla*$_{NDM-5}$, *bla*$_{TEM-1}$, and *sul1*, *aadA2* and *dfrA12* in a class 1 integron. Although the *bla*$_{NDM-5}$ gene was also present in some B4/H24RxC isolates, in those it was found in variants of the *bla*$_{OXA-181}$ carrying X3 plasmid (Fig. S6) as previously reported[16,17].

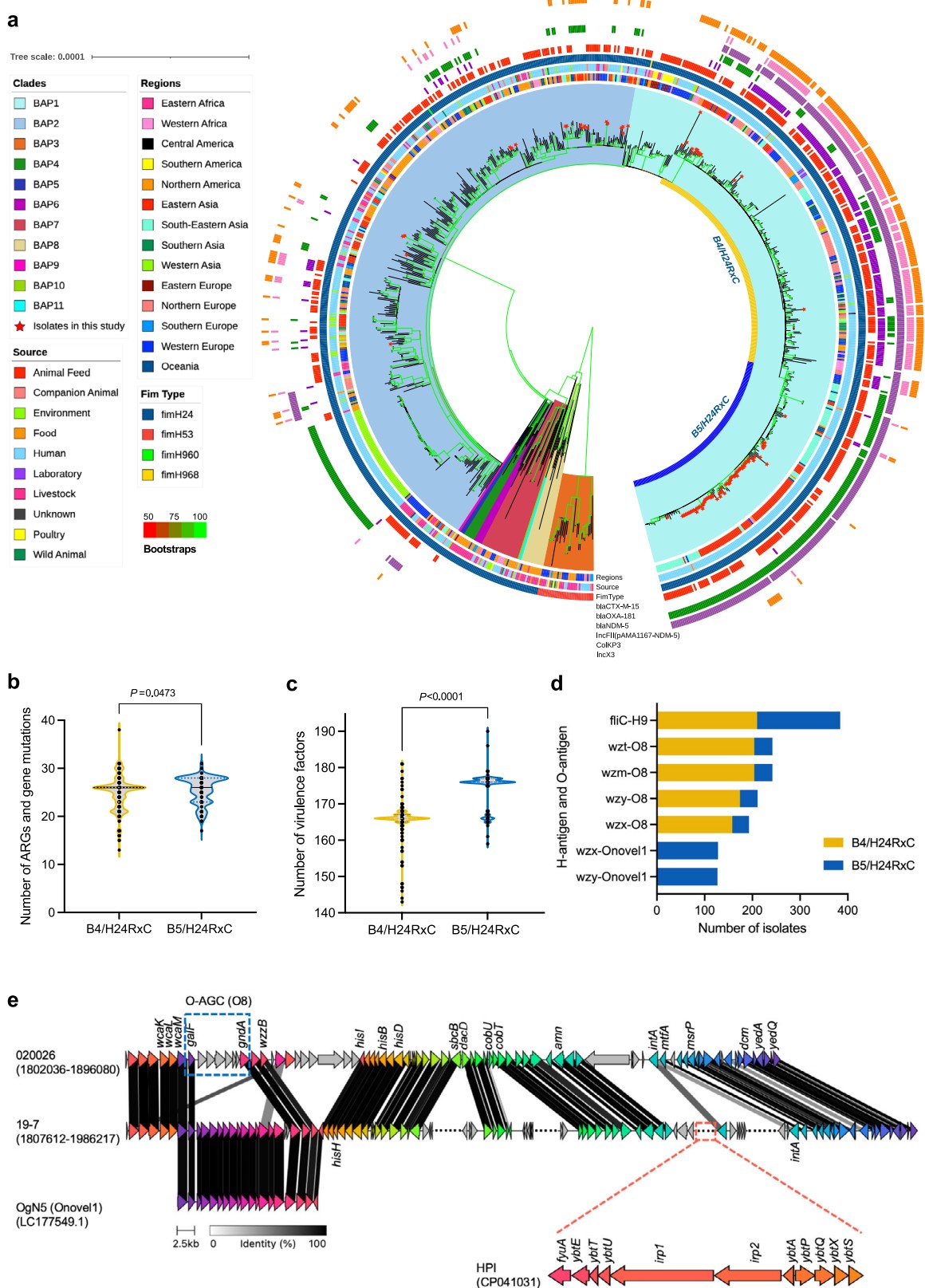

## Core-genome genes and SNPs associated with B5/H24RxC MDR clone

Pangenome-wide association study (pan-GWAS) using Scoary for the two clones identified 199 genes that were positively or negatively associated with B5/H24RxC (Fig. 5; Supplementary Data 6). Amongst these genes, 84 were annotated as non-hypothetical, of which 44 were over-represented in B5/H24RxC and 40 under-represented. In addition to the expected identification of accessory genes carried by different plasmids in these two clones, chromosomal accessory genes were also found to be associated with B5/H24RxC. A number of these chromosomal genes, including all 11 genes located in the HPI, were found in the majority of B5/H24RxC isolates (75.3–77.6%, 131–135/174), but in none

**Fig. 3 | Phylogeny of a global ST410 collection. a** Midpoint rooted maximum-likelihood phylogeny of 956 global ST410 was constructed using a core-genome SNP alignment generated by Snippy v4.6.0 with ST410 isolate YD786 (GenBank accession CP013112.1) as the reference. Branch support was performed with 1000 bootstrap replicates. Bootstrap values are represented with gradient colours. Isolates from this study are indicated with a red star. **b** Violin plot showing the distribution of total number of ARGs and mutations that confer resistance in B4/H24RxC and B5/H24RxC clones. Statistical difference between the two clones was assessed with two-tailed unpaired Student's *t* test. **c** Violin plot showing distribution of total number of virulence factors in B4/H24RxC and B5/H24RxC clones. Statistical difference was assessed with two-tailed unpaired Student's *t* test. **d** Bar plot showing the presence of the lipopolysaccharide (O) and flagellar (H) surface antigens in B4/H24RxC and B5/H24RxC clones. **e** Comparison of the recombination regions in strain 020026 and 19-7 identified the O-antigen switch from O8 in B4/H24RxC to Onovel1 (OgN5) in B5/H24RxC and the HPI gene cluster in B5/H24RxC clone. Source data are provided as a Source Data file.

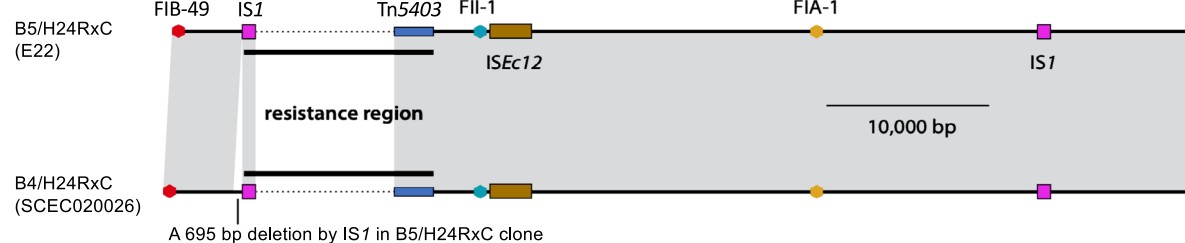

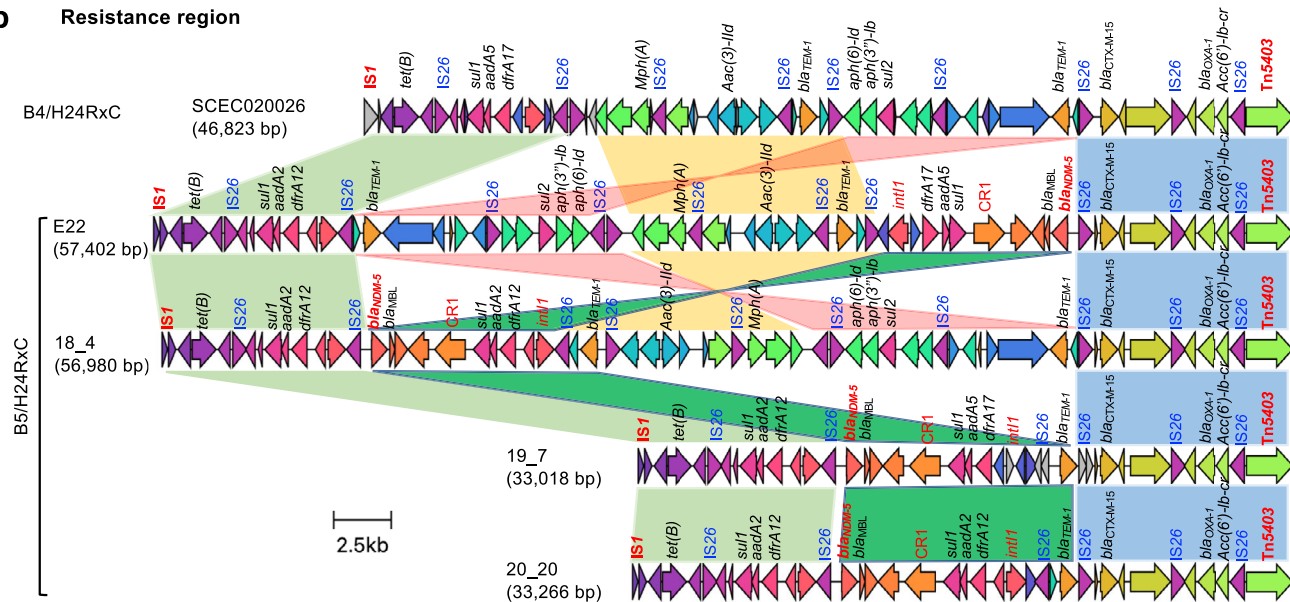

**Fig. 4 | FII-1:FIA-1:FIB-49 plasmids analysis. a** Comparison of the backbone of the F-type plasmids in the B4/H24RxC and B5/H24RxC clones. **b** Comparison of resistance regions found in F-type plasmids. Genbank accessions for plasmids: pCTXM15_020026 (CP034956), pE22P1 (CP123037), p18-4P1 (CP123014), p19-7P2 (CP123019), p20-20P2 (CP123031).

---

of the B4/H24RxC isolates. A prophage integrase gene (*intA*), a histone-like protein gene (*hns*) and two 2,3,4,5-tetrahydropyridine-2,6-dicarboxylate N-acetyltransferase genes (*dapH*) were also found in B5/H24RxC (75.3–77.6%, 131–135/174) but not in B4/H24RxC. Consistent with the serotyping result, O-antigen genes *wzm/wzt* O8 were associated with B4/H24RxC (204/214, 95.3%) and *wzx/wzy* Onovel1 were associated with B5/H24RxC (130/174, 74.7%) (Fig. 5).

Similarly, association analysis using Scoary revealed that 423 chromosomal SNPs were positively associated with B5/H24RxC (Supplementary Data 7). Amongst the 423 SNPs, 403 were in coding sequences and 20 were in intergenic regions. Only 63 of the 403 substitutions in coding sequences were non-synonymous, present in 32 genes (Fig. S7; Supplementary Data 7), most of which were found in the recombination regions, such as genes in the histidine operon (*hisB, hisD* etc.), the colanic acid biosynthesis genes cluster (*wcaL* and *wcaM*) and the nitrogen assimilation transcriptional regulator gene *nac*. Premature stop codons were found in three genes which encode a PstS

family phosphate ABC transporter substrate-binding protein, the inverse autotransporter adhesin-like protein YeeJ and the cardiolipin synthase ClsB. 145 SNPs were exclusively found in B5/H24RxC, with only 12 present in all 174 isolates investigated. Gubbins revealed that amongst the B5/H24RxC specific SNPs, 129 were found to be introduced via recombination events (Fig. S7; Supplementary Data 7).

**Time of origin of the B5/H24RxC MDR clone**
The estimated time of the most recent common ancestor (TMRCA) of different phylogenetic groups was investigated with BEAST2 on the 500 Treemmer selected ST410 genomes from the 956 global collection. A mutation rate of 6.42E-7 SNPs per site per year [95% highest posterior density (HPD) intervals 5.69E-7, 7.19E-7] was estimated. The analysis estimated the age of the ST410 lineage to be approximately 205 years, with a TMRCA of around 1816 (95% HPD, 1739–1879) (Fig. 6a), close to a previous estimate of 1803[15]. The B4/H24RxC ancestor was estimated to have originated in 2003 (95% HPD,

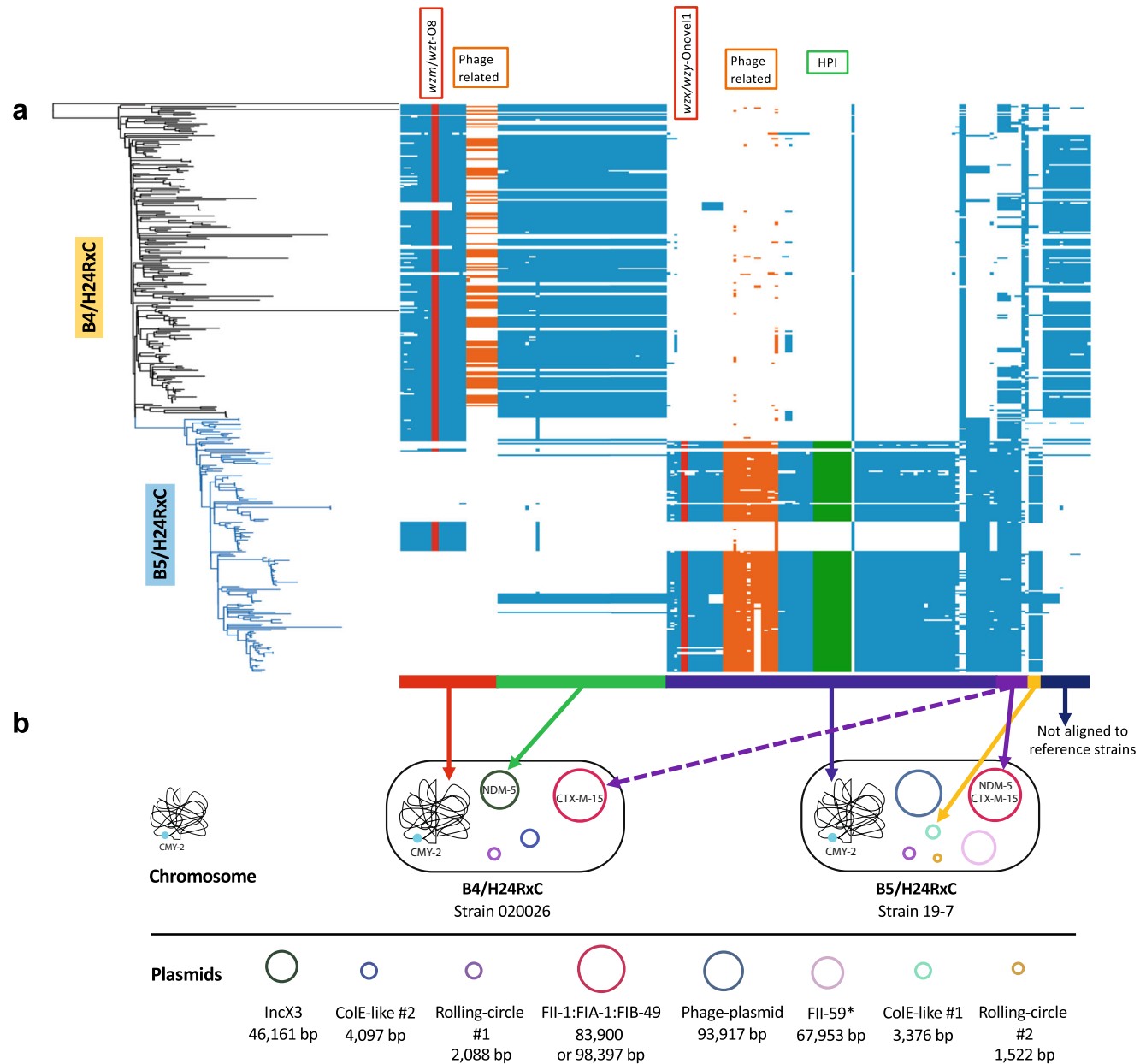

**Fig. 5 | Core-genome genes associated with the B5/H24RxC MDR clone.**
**a** Presence and absence of the genes positively and negatively associated with the B5/H24RxC clone mapped to the phylogeny. Genes are aligned and ordered against the complete genomes of the reference strains 020026 (Genbank: CP034954 to CP034958) and 19-7 (CP123017 to CP123023). Genes located in phage regions are shaded in orange, genes found in the high pathogenicity island (HPI) are in green and O-group genes are in red. **b** Schematic representation of the chromosome and plasmids in the reference strains 020026 and 19-7. Colour bars and arrows indicate the location of the genes in the genomes of the reference strains. $bla_{CMY-2}$ was chromosomally integrated in B4/H24RxC as reported previously[17] and in B5/H24RxC. Source data are provided as a Source Data file.

2000–2005), which is identical to the estimate of 2003 from the same previous study[15]. The TMRCA of B5/H24RxC was estimated at around May 2006 (95% HPD, 2004–2008) (Fig. 6c).

**B5/H24RxC shows a fitness advantage and enhanced virulence**
In two different assays, isolates of the B5/H24RxC clone exhibited a fitness advantage compared with isolates of the B4/H24RxC clone in LB medium. Growth curve analysis showed that the isolates of B5/H24RxC grew considerably better, with a reduced doubling time of 68.5 mins in half strength LB (average of 6 isolates) compared with isolates of B4/H24RxC (83.5 mins, average of 2 isolates) (Fig. 7a, b; Table S2). In full strength LB and 1/10 strength LB, B5/H24RxC isolates also showed significantly better growth (Fig. S8; Table S2). Genomic

analysis revealed that these isolates of the B5/H24RxC clone possessed gene *fyuA* but not *yodB*, while isolates of the B4/H24RxC clone contained gene *yodB* but not *fyuA*. A qPCR-based (targeting *yodB* and *fyuA*) growth competition experiment between isolates of the two clones also confirmed that isolates of B5/H24RxC grew faster than isolates of B4/H24RxC, as demonstrated by the increasing ratio of *fyuA* to *yodB* levels in the same culture (Fig. 7c, d).

In a wax moth larvae infection model, larvae infected with isolates of B5/H24RxC had a significantly lower survival rate ($P < 0.002$ or $P < 0.001$, Log-rank [Mantel−Cox] test) than those infected with isolates of B4/H24RxC (Fig. 7e; Table S3) after 72 h. The result also showed that larvae infected with isolates of B5/H24RxC had a lower survival rate than those infected with hypervirulent *Klebsiella pneumoniae*

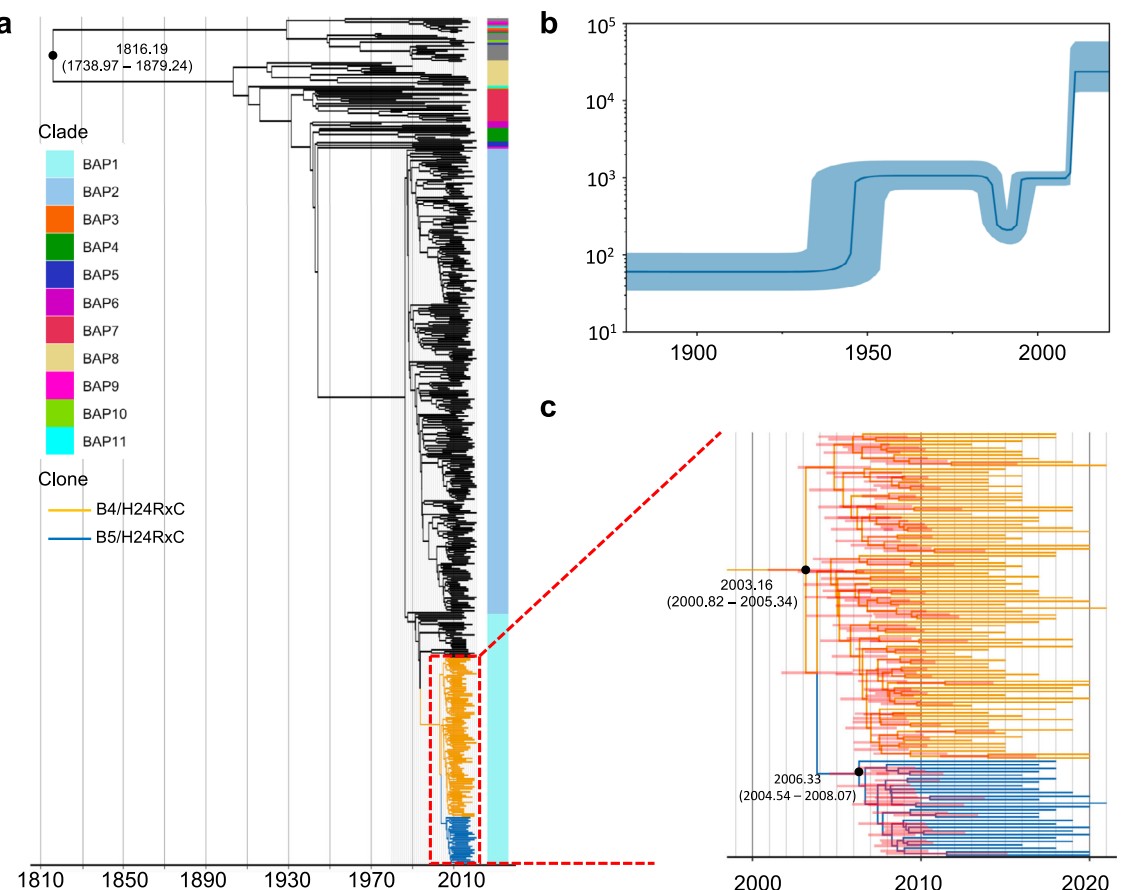

**Fig. 6 | Coalescence-based analysis of *E. coli* ST410. a** A time-calibrated phylogeny was reconstructed using BEAST2.0 based on the nonrecombinant SNPs for the 500 selected *E. coli* ST410. The MDR clone B4/H24RxC and B5/H24RxC are coloured in orange and blue, respectively. **b** The Bayesian skyline plot illustrates the predicted demographic changes of the ST410 clades. The thick solid line represents the median estimate of the effective population size, with 95% confidence interval shown in lighter blue area. **c** An enlarged phylogenetic tree showing the B4/H24RxC and B5/H24/RxC clones.

strain K1088 ($P = 0.0053$ to $P = 0.0263$, Log-rank [Mantel–Cox] test) and a similar survival rate to those infected with hypervirulent *Acinetobacter baumannii* strain AB5075 ($P > 0.1$, Log-rank [Mantel–Cox] test). Based on this data relative to these two hypervirulent bacterial strains, we consider that B5/H24RxC is a hypervirulent *E. coli* clone. This result was expected as these B5/H24RxC isolates possessed the characterised pathogenicity island HPI, which was not present in the B4/H24RxC isolates. The HPI also contributes to iron acquisition by *E. coli* through the production of yersiniabactin[29]. Our iron source growth assay confirmed the enhanced ability of B5/H24RxC to utilise iron in comparison to B4/H24RxC (Fig. 7f). The growth of all tested isolates was completely inhibited in the presence of 300 µM DIP, suggesting that they were equally resistant to iron-deprived conditions. The addition of $FeCl_2$ restored growth of all isolates, while the addition of holo-transferrin only restored the growth of B5/H24RxC isolates. The result also showed that a higher concentration of haemoglobin and hemin was required to restore growth of the B4/H24RxC isolates than the B5/H24RxC isolates. Interestingly, growth was restored in none of the isolates with the addition of lactoferrin, which was different from a previous study where B4/H24RxC isolates tested were shown to be able to utilise iron from lactoferrin[16]. It suggests that not all B4/H24RxC isolates could utilise iron from lactoferrin as we had used two different isolates in this study.

Biofilm formation assays revealed that isolates of B4/H24RxC clone were poor biofilm formers as suggested previously[16]. The biofilm formation abilities of isolates of the B5/H24RxC clone were also categorised as weak or non-production (Fig. S9) following previously proposed interpretation criteria[30,31].

### B5/H24RxC is globally disseminated

The isolates of the B5/H24RxC clone ($n = 174$) in this international collection ($n = 956$) were collected between 2015 and 2021 from humans, food and companion animals across 11 countries on 5 continents (Fig. S1; Supplementary Data 2). The majority of the isolates (89.1%, 155/174) were from countries in eastern and south-eastern Asia, such as China ($n = 78$) and Thailand ($n = 71$).

To capture a more up-to-date picture of the spread of the B5/H24RxC clone worldwide since our analysis, we screened a new collection of *E. coli* ST410 genomes ($n = 714$, Supplementary Data 8) available from 14 Jan 2022 to 27 Sept 2023 on EnteroBase. A further 84 B5/H24RxC genomes were identified from the collection (Fig. S10; Supplementary Data 8). The isolates providing these genomes were collected between 2016 and 2023 across 13 countries on 5 continents (Fig. S10), mainly from humans, with two isolates from dogs and two from environmental samples. The countries with the most isolates were Thailand ($n = 26$), United States ($n = 17$), Germany ($n = 7$), China ($n = 6$), Cambodia ($n = 6$), Australia ($n = 5$) and Norway ($n = 4$), suggesting the ongoing spread of this clone globally.

## Discussion

In this study, we report the emergence of a hypervirulent MDR clone of *E. coli* ST410. *E. coli* ST410 has emerged recently as a successful MDR extraintestinal pathogenic *E. coli* lineage responsible for increasing numbers of infections worldwide[15,16]. Our genomic surveillance of CREC in Chinese hospitals between 2017 and 2021 revealed that ST410 was the most commonly isolated CREC ST, overtaking ST167, ST131

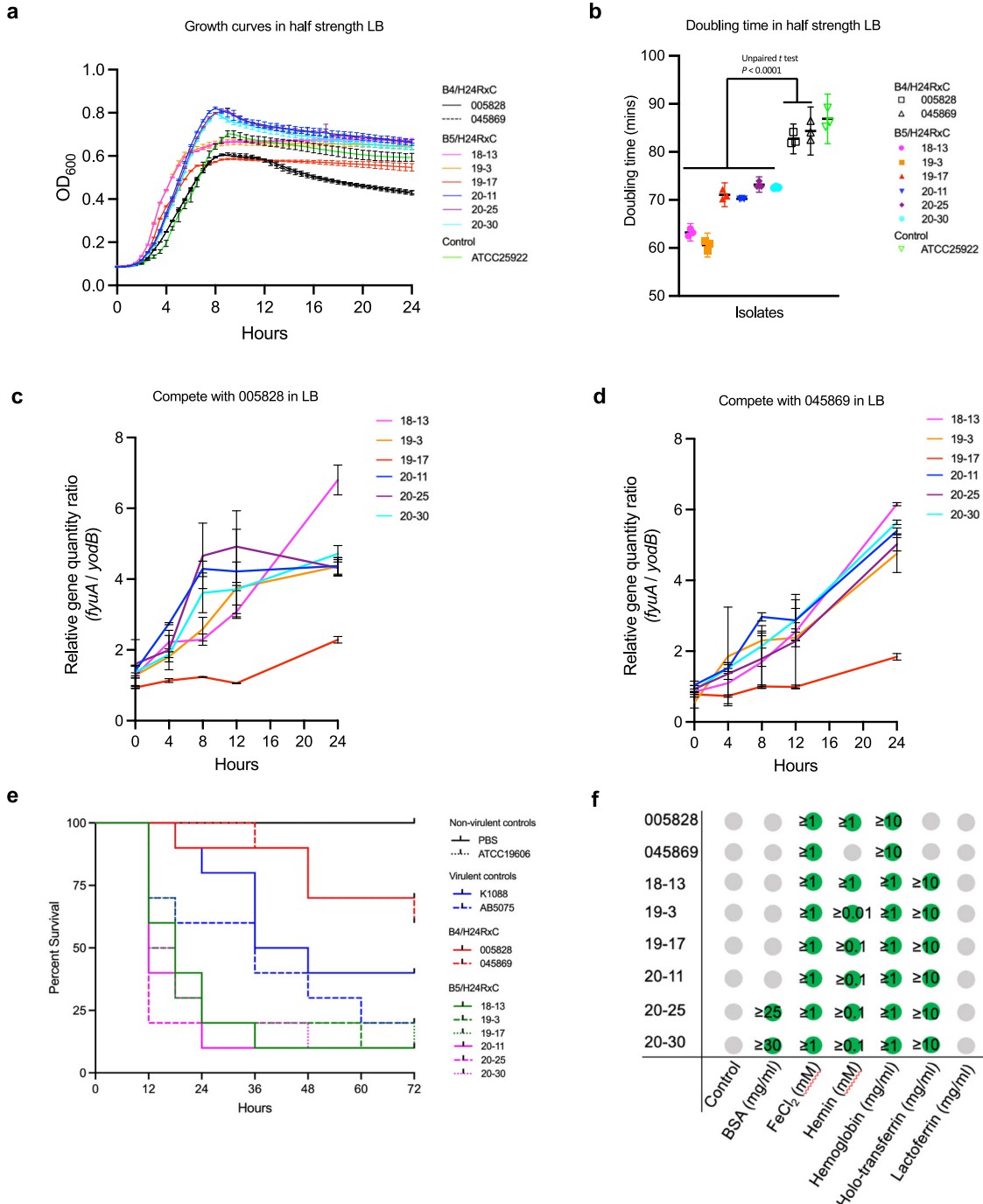

**Fig. 7 | Phenotypic comparison of B4/H24RxC and B5/H24RxC clones. a** Growth curves in half strength LB for strains of both clones. Strain ATCC 25922 was included as a growth control. Data are shown as mean ± SD from $n = 3$ biological replicates. **b** Doubling time in half strength LB for isolates of both clones. Strain ATCC 25922 was included as a growth control. Data are shown as mean ± SD from $n = 3$ biological replicates. Statistical difference was assessed with two-tailed unpaired Student's $t$ test. **c, d** qPCR-based competition assay for strains of both clones. The figures show the relative quantity ratio of gene *fyuA* in B5/H24RxC strains to gene *yodB* in B4/H24RxC strains 005828 and 045869. Data are shown as mean ± SD from $n = 3$ biological replicates. Statistical difference was assessed with two-tailed unpaired Student's $t$ test. **e** Survival curves for wax moth larvae (*G.*

*mellonella*) infected with ~2 × 10⁶ CFU of different isolates in the B4/H24RxC and B5/ H24RxC clones. Hypervirulent *Klebsiella pneumoniae* strain K1088[65] and hyper-virulent *Acinetobacter baumannii* strain AB5075[66] were used as positive controls while *Acinetobacter baumannii* ATCC 19606 and PBS were used as negative controls. The curves represent the mean of three biological repeats. **f** The ability to utilise different iron sources by B4/H24RxC and B5/H24RxC clones. Green circles indicate growth and the numbers inside show the lowest tested concentration of the iron sources needed for the isolates to grow. Grey circles indicate no growth was observed at any concentration of the iron sources used. $n = 2$ biological independent experiments with the same result. Source data are provided as a Source Data file.

and ST617 which were reported to be most common during surveillance carried out in 2015–2017[9,14]. Previous studies have identified significant clades in the ST410 lineages based on their genomic characteristics, and reported a globally widespread MDR clone, B4/

H24RxC[15,16]. The clone identified here, B5/H24RxC, possessed distinctive features that distinguish it from B4/H24RxC. It was similarly globally disseminated and was found to have caused two separate outbreaks in a children's hospital in eastern China.

Genomic analysis revealed that the B5/H24RxC clone was closely related to and likely emerged from B4/H24RxC. Both clones were grouped into BAP1 by Fastbaps[32] based on a SNP alignment and a phylogeny of an international collection of ST410 genomes ($n = 956$). However, the B5/H24RxC clone lacked the X3 plasmid-borne $bla_{OXA-181}$, which is a defining feature of B4/H24RxC[15]. Instead, B5/H24RxC carried $bla_{NDM-5}$ in a F-type plasmid derived from an ancestral variant found in B4/H24RxC, which lacks $bla_{NDM-5}$. However, $bla_{NDM-5}$-bearing plasmids of the same F-type lineage, F1:A1:B49, have been described in some B4/H24RxC isolates[17], suggesting either that this plasmid lineage acquired $bla_{NDM-5}$ in a B4/H24RxC host that was the progenitor to B5/H24RxC, or that the IS26-associated $bla_{NDM-5}$ was acquired on at least two occasions. Importantly, the majority of B5/H24RxC isolates investigated in this study carried the high pathogenicity island HPI and a novel O-antigen (Onovel1) which were likely introduced via recombination (Fig. S2). Based on these findings, we argue that adding a new sub-lineage (B5/H24RxC) to the Roer et al. [15] classification scheme provides better resolution for distinguishing ST410 *E. coli* with distinctive genotypic and phenotypic features than the modified classification scheme proposed by Chen et al. [17].

Our wax larvae infection assay demonstrated that the HPI-containing B5/H24RxC was more virulent than B4/H24RxC, adding further evidence that HPI contributes to enhanced virulence in *E. coli* as previously described[29,33]. The HPI element is frequently found in the genomes of extraintestinal pathogenic *E.coli* (ExPEC) associated with UTIs[34], especially in uropathogenic *E.coli* (UPEC) strains[35,36]. The HPI allows *E. coli* to obtain iron from the urinary tract, which is normally a low-iron environment. This enhances the ability of the bacteria to colonise and persist within the urinary tract and causing infections[37,38]. Our data from the children's hospital in eastern China also showed that a large proportion of HPI-carrying B5/H24RxC isolates were from patients with UTI, suggesting that HPI may have played a role in the emergence of this highly virulent clone. However, it should be noted that we did not explore other genetic factors, apart from the HPI, that could contribute to the enhanced virulence in the B5/H24RxC clone. Another important genomic difference that may also have contributed to the emergence and dissemination of B5/H24RxC is the O-antigen switching from O8 in B4/H24RxC to Onovel1 in B5/H24RxC. It is not clear why *E. coli* changes its O-antigen, but it is known that the ability to vary the O-antigen structure is important for bacterial adaptation to changing environments, including evasion of host immune responses[39,40]. Our data also indicated that B5/H24RxC had a fitness advantage over B4/H24RxC, exhibiting a higher growth rate in vitro. These traits could explain the successful global dissemination of B5/H24RxC over the last 15 years.

Additionally, non-synonymous SNPs associated with B5/H24RxC clone were identified in 32 genes, which are mostly involved in metabolism and cell wall biosynthesis, such as the genes in the histidine operon and the colanic acid biosynthesis genes cluster. It is unclear whether these substitutions in these genes contributed to the emergence of B5/H24RxC from B4/H24RxC, but they may have played a role in its increased fitness. Interestingly, the inverse autotransporter adhesin-like protein YeeJ, which was reported to promote biofilm formation[41], was disrupted by a premature stop codon caused by an SNP. This may have contributed to the poor ability of B5/H24RxC to form biofilm (Fig. S9), although B4/H24RxC with an intact YeeJ is also a poor biofilm former[16]. It should also be noted that, in this study, non-synonymous SNPs associated with B5/H24RxC were not further investigated to validate their possible links to some phenotypic changes. Therefore, it remains unclear if YeeJ plays a role in biofilm formation in these CREC clones.

Coalescence analysis on a global selection of 500 ST410 isolates indicated a most recent comment ancestor existed ~205 years ago (1816; 95% HPD, 1739–1879), similar to a previous estimate of 1803[15]. The B4/H24RxC clone was estimated to have originated in 2003, which

also agrees with the same previous study[15]. The TMRCA of the B5/H24RxC clone was estimated at around May 2008 (95% HPD, 2004–2008).

In this study, we have identified a CREC ST410 clone, B5/H24RxC, that may have greater clinical impact relative to its precursors due to the acquisition of resistance elements and the high pathogenicity island, HPI. Although the majority of B5/H24RxC were of human origin because of sampling bias, a few isolates ($n = 3$) were collected from companion animals and food samples, implying that various routes of transmission may be contributing to its dissemination. It is important to note that at the time of this study, most B5/H24RxC isolates were collected from low- and middle-income countries. The multidrug-resistant and hypervirulent nature of this clone, coupled with various socioeconomic factors in these countries, makes it a challenging problem for healthcare systems already stretched to their limits. The emergence of both B4/H24RxC and B5/H24RxC MDR clones over the last two decades highlights the rapidly evolving landscape of pathogenic *E. coli*, which is being driven by continued evolution towards enhanced resistance and virulence, which in turn is being driven by recombination of key loci involved in mammalian pathogenesis and colonisation. To mitigate the impact of newly emerged and future clones, additional research is required to understand the evolutionary mechanisms involved in the emergence of the new *E. coli* clones that are of human and veterinary clinical importance.

## Methods

### Ethics and consent
This study was approved by the medical ethics committee of the First Affiliated Hospital of Guangzhou Medical University (GMU) on 21 May 2018. For the work involving the CREC isolates from the children's hospital, ethics was approved by the medical ethics committee of the Children's Hospital of Soochow University on 5 January, 2021. Individual consent was obtained from the patients' guardians by hospital staff.

### Bacterial isolates and antimicrobial susceptibility testing
An initial collection of 168 CREC isolates (GMU collection) were isolated from patients admitted to municipal hospitals in Guangzhou, Jiangsu and Beijing during 2018–2021. All isolates were identified as *E. coli* by a VITEK 2 Compact system (BioMérieux, Marcy l'Etoile, France) and confirmed to be carbapenem resistant by antimicrobial susceptibility testing. Minimum inhibitory concentrations (MICs) of meropenem, imipenem and ertapenem were determined by broth microdilution method according to the Clinical Laboratory and Standards Institute (CLSI) M100 (31st edition). The MICs of other antibiotics, including polymyxin B and tigecycline, were also determined by broth microdilution method according to European Committee on Antimicrobial Susceptibility Testing (EUCAST). For polymyxin B and tigecycline, the breakpoints defined by the EUCAST were used. All antibiotics were purchased from Macklin, Shanghai, China. *E. coli* ATCC 25922 and *P. aeruginosa* ATCC 27853 were used for standardisation.

A further 220 CREC collected by the China Antimicrobial Surveillance Network (CHINET) from Chinese hospitals in 26 provinces during 2017–2021 were also included in this study. Whole-genome sequencing data and relevant metadata of the isolates, including MICs for meropenem, imipenem, polymyxin B and tigecycline, were provided by CHINET. Metadata for isolates from both collections are presented in Supplementary Data 1.

### Whole-genome sequencing and publicly available sequences
Whole-genome sequencing was performed for all CREC isolates ($n = 168$) for the initial collection. Briefly, single colonies from an overnight agar plate were cultured in 4 ml of LB broth at 37 °C for 16 h, and genomic DNA was extracted using a Bacterial DNA Kit D3350

(Omega BioTek, USA). Sequencing was conducted by Novogene (Beijing, China) using an Illumina Novaseq 6000 platform (Illumina, San Diego, CA, USA). Five isolates were also subjected to whole-genome sequencing using the long-read MinION Sequencer (Nanopore; Oxford, UK) by Novogene (Beijing, China). These five isolates (E22, 18-4, 20-16, 19-7, and 20-20; Table S1) belong to the B5/H24RxC clone and they were selected based on their genomic differences, representing the range of diversity as suggested by their positions in the phylogeny.

For the analysis of a resistant clone of *E. coli* ST410 identified in this study, EnteroBase (http://enterobase.warwick.ac.uk/) was searched for *E. coli* ST410 genomes (accessed on 13 Jan 2022) and only genomes with relevant metadata (year of collection, country and source type) and availability of raw reads were included (*n* = 790). *E. coli* ST410 genomes used by a previous study[15] that were not present in the EnteroBase data set (*n* = 84) were also included. SRA and ENA accession numbers were extracted and raw reads for the genomes were downloaded from the European Nucleotide Archive. Metadata for the genomes are presented in Supplementary Data 2.

### Genome assembly, annotation, and characterisation
Illumina sequence reads were trimmed and assembled with Shovill v1.1.0 with default settings (https://github.com/tseemann/shovill) and assemblies were assessed for contamination and completeness using QUAST v5.0.2, CheckM v1.1.3 and Centrifuge v1.0.4[42–44]. Assemblies with a genome size greater than six million base pairs, an N50 smaller than 15,000 or a genome contamination greater than 2% were excluded from further analysis. A de novo hybrid of assembly of both Illumina reads and Nanopore reads was carried out using Unicycler v0.4.8 with default settings[45]. Prokka v1.14.0 was used to annotate the genome sequences[46]. Acquired ARGs and mutations that confer resistance were identified using abritAMR v1.0.13, an ISO-certified bioinformatics platform for genomics-based bacterial AMR detection[47]. Plasmid replicons, virulence factors and serotypes were identified with ABRicate v1.0.1 (https://github.com/tseemann/abricate), using the Resfinder[48], Plasmidfinder[49], Ecoli-vf and EcOH[27] databases (updated on 15 September 2022) with default parameters. Multilocus sequence types (MLSTs) were determined using MLST v2.19.0 (https://github.com/tseemann/mlst) with the "ecoli" scheme. *fimH* types were identified with FimTyper v1.0[50].

Protein sequences of GyrA, GyrB, ParC, ParE, FtsI (PBP3), and FyuA were obtained from all genomes by performing a TBLASTN query of a representative protein sequence for each of the six proteins. Amino acid substitutions in the quinolone resistance-determining regions (QRDRs; in GyrA, GyrB, ParC and ParE) and insertion in FstI were identified by comparing their protein sequences with the corresponding protein sequences from quinolone- and fluoroquinolone-susceptible *E. coli* K-12 MG1655 GenBank: NC_000913.3).

### Phylogenetic analysis of *E. coli* ST410
To construct a global phylogeny for ST410 *E. coli*, 956 genomes (109 from this study and 847 publicly available genomes that passed quality control) were used to generate a full-length whole-genome alignment using Snippy v4.6.0 (https://github.com/tseemann/snippy) with ST410 isolate YD786 (GenBank accession CP013112.1) as the reference. The full-length whole-genome alignment was cleaned with the snippy-clean function and then used as an input to Gubbins v2.4.1[51] for identifying and filtering regions of homologous recombination. Variant sites in the alignment were extracted using SNP-sites v2.5.1[52] and a maximum-likelihood (ML) phylogenetic tree was reconstructed using IQ-tree[53] accounting for constant sites in the alignment and run for 1000 bootstraps using the extended model selection function. The resulting tree was annotated in iTOL[54]. The phylogeny of the ST410 CREC isolated from a Chinese children's hospital with potential outbreaks was

also constructed with the complete genome of isolate 19-7 as the reference using the methods described above.

Analysis of population structure within the ML phylogeny was conducted using Fastbaps v1.0 to identify major clusters under default parameters[32]. Pairwise SNP distances between isolate genomes were calculated using snp-dists v0.8.2 (https://github.com/tseemann/snp-dists) using the recombination removed alignment from Gubbins as input.

### Comparative genomic analysis of B4/H24RxC and B5/H24RxC clones
Prokka-annotated sequences of both B4/H24RxC (*n* = 214) and B5/H24RxC (*n* = 174) were analysed with Panaroo v1.2.7[55] under default settings to infer core and pangenomes. Clone association analysis of the genes in the pangenome matrix was performed with Scoary v1.6.16[56] with a maximum Benjamini–Hochberg adjusted *P* value of 1E-30.

Using the method described above, a smaller scale core-genome SNP phylogeny was reconstructed only for the genomes in these two clones with the complete genome of a previously reported isolate 020026 (B4/H24RxC; Genbank: CP034954 to CP034958) used as the reference[16] and isolate A8 (B3/H24Rx) from this study as outgroup. SNPs specific to the B5/H24RxC MDR clone were identified by feeding the entire SNP matrix of all genomes of both clones into Scoary v1.6.16[56] using the same settings as described above. Gubbins v2.4.1[51] was used to determine whether the clone-specific genes and SNPs were due to recombination. The exact sequence locations for phage regions in the complete genomes of reference strains 19-7 and 020026 were identified using the PHASTER server[57].

Plasmid sequences were examined and manually annotated in Gene Construction Kit v4.5.1 (Textco Biosoftware, Raleigh, USA). Plasmid replicons were initially identified with the PlasmidFinder database as described above, and PubMLST was used to sub-type F-type replicons (https://pubmlst.org/organisms/plasmid-mlst). Insertion sequences were identified using the ISFinder database (https://isfinder.biotoul.fr/).

The sequence of the O-antigen Onovel1 identified in the majority of B5/H24RxC isolates was used to perform a web nucleotide BLAST. The top match was a previously reported O-antigen OgN5[27]. Pairwise alignment and visualisation of the two sequences were carried out using clinker v0.0.28[58].

### Coalescent analysis of *E. coli* ST410
Treemmer v0.3[59] was used to reduce the ST410 global phylogeny to 500 genomes, maintaining ~95% of the original genetic diversity. The subsampled genomes had also maintained the diversity in year of isolation and country of origin (Fig. S4). A time-calibrated phylogeny for ST410 was reconstructed using the SNP alignment data generated from the raw sequence data of the selected 500 genomes as described above. Each sequence in the alignment was annotated with the year of isolation. The presence of a temporal signal in the data was investigated by inferring linear relationship between root-to-tip distances of the phylogenetic branches and the year of sample isolation using TempEst v1.5.3[60], which revealed a correlation coefficient of $R^2$ = 0.32. Coalescence-based analysis was performed with BEAST v2.6.6[61]. In order to identify the most suitable model, analyses were performed using different substitution models (GTR and HKY), strict and relaxed molecular clock, and different demographic models including Bayesian Skyline, constant population and exponential population. Model selection was performed using Nested Sampling[62] v1.1.0 within the BEAST2 package with a particle count of 1, sub chain length of 5000, and Epsilon of $1.0 \times 10^{-12}$. The best fit model was estimated to be a GTR model and a relaxed molecular clock with a Bayesian Skyline population model (Table S4). Three replicates for this model were run for 700 million Markov chain Monte Carlo (MCMC) iterations, sampling every

50,000 states. Log files were combined with a 10% burn-in using LogCombiner v2.6.6 and assessed for convergence by checking the effective sample size (ESS, >200 for each parameter) using Tracer v1.7.2. A maximum clade creditability tree summarising the posterior sample of trees in the combined MCMC runs was generated with TreeAnnotator v2.6.6. The resulting tree was annotated and visualised with iTOL[54] and ggtree v3.2.1[63].

## Growth curves and qPCR-based competition assays

An optical growth analyser (BioTek Epoch2, USA) was used to monitor the growth rate of ST410 strains of B4/H24RxC and B5/H24RxC clones. Based on their availability and their genetic characteristics, B4/H24RxC isolates (005828 and 045869) and B5/H24RxC isolates (18-13, 19-3, 19-17, 20-11, 20-25, and 20-30) were selected for the assay. Briefly, overnight cultures in LB were adjusted to a turbidity equivalent to that of a 0.5 McFarland standard and then inoculated 1:1000 in fresh LB, half strength LB or 1/10 strength LB. For each strain, 300 µl of inoculated medium was added into wells of the microplate in triplicate. Fresh medium was also added to three wells acting as blank controls. Cultures were incubated at 37 °C with continuous shaking for 24 h and $OD_{600}$ was measured every 30 min. Growth curves for each strain during the exponential phase were analysed with the GraphPad Prism 9 software. The doubling time was calculated from growth curve using fitted curve of sigmoid function with a Python script available at https://github.com/huoww07/calulate_bacteria_doubling_time. Strain ATCC 25922 was included as a growth control in the assay. Two-tailed unpaired Student's $t$ test was employed to measure the difference between the doubling time of the two clones.

The growth dynamics of both clones were further investigated using a qPCR-based head to head competition assay in LB. In each competition group, a *yodB* positive isolate of the B4/H24RxC clone was in competition with a *fyuA* positive isolate of the B5/H24RxC clone. For instance, overnight cultures of isolate 005828 (B4/H24RxC) and 18-13 (B5/H24RxC) in LB were adjusted to a turbidity equivalent to that of a 0.5 McFarland standard and equal amount of the adjusted cultures were inoculated 1:200 into the same tube of fresh LB. Culture was incubated at 37 °C with continuous shaking for 24 h and samples were taken at hour 0, 4, 8, 12, and 24 for DNA extraction using the Bacterial DNA kit D3350 (Omega BioTek, USA). qPCR was performed using the HiScript III RT SuperMix for qPCR kit (Vazyme, China) on a Roche LightCycler PCR instrument (Roche, Sweden) using *yodB* and *fyuA* primers. Relative amount of the genes was calculated using 16S rRNA as the reference. The $2^{-\Delta\Delta Ct}$ method was used to determine the ratio of *fyuA* to *yodB*. The primers used were 16srRNA-F: GGAAGAAGCTTGCT TCTTTGCTGAC and 16srRNA-R: AGCCCGGGGATTTCACATCTGACT TA; *yodB*-F: GGTGGCACAATAGAAGGAT and *yodB*-R: TTATCTGCT-GATGCGAGAA; *fyuA*-F: CAGTAGGCACGATGTTGTA and *fyuA*-R: GCTATCCGCAGGCTATATG.

## Virulence assay

The wax moth (*Galleria mellonella*) larvae assay was based on a previously described method[16,64]. The larvae (~35 days after hatching and of 300 ± 50 mg) were purchased from Huiyude Biotechnology (Tianjin, China). B4/H24RxC isolates (005828 and 045869) and B5/H24RxC isolates (18-13, 19-3, 19-17, 20-11, 20-25 and 20-30) were assessed for their virulence. Hypervirulent *Klebsiella pneumoniae* strain K1088[65] and hypervirulent *Acinetobacter baumannii* strain AB5075[66] were used as positive controls while *Acinetobacter baumannii* ATCC 19606 and PBS were used as negative controls. Single bacterial colonies were used to inoculate 5 ml of LB and incubated at 37 °C with 200 rpm shaking for about 6 to 8 h to reach the exponential phase. Cultures then were adjusted to a turbidity equivalent to $10^8$ CFU/ml in PBS as predetermined by plate count and further diluted 1:10 to obtain cultures of $10^7$ CFU/ml. For each strain, three groups of larvae ($n = 10$ in each

group) were injected with 20 µl of aliquots of diluted culture via the last left proleg using a Disposable Sterile Insulin Syringe U-100 (B.Braun, Germany). The infected larvae were kept at 37 °C during the course of the assay and survival of the larvae was recorded at hour 6, 12, 18, 24, 36, 48 and 72. Survival curves were generated and analysed using GraphPad Prism 9 software.

## Iron source growth assay

The growth of B5/H24RxC isolates (18-13, 19-3, 19-17, 20-11, 20-25, and 20-30) were evaluated under different iron sources on LB agar plates using B4/H24RxC isolates (005828 and 045869) as controls as described previously with modifications[16,67]. Briefly, the MIC of 2′2-dipyridyl (DIP; Macklin, China) was determined using a previously described method[16] and all the isolates had an MIC of 300 µM. For each isolate, a single colony from an overnight LB agar plate was cultured in LB broth containing 200 µM DIP for 6 h to limit the growth of the isolate. Bacterial cells were then collected by centrifugation at $4000 \times g$, washed and resuspended in PBS to a turbidity equivalent to that of a 0.5 McFarland standard. About $10^5$ CFU of each isolate were then spread onto LB agar plates containing DIP at the MIC (300 µM for all isolates). Iron sources (10 µl) of different concentrations including bovine serum albumin (10 mg/ml, 25 mg/ml, 50 mg/ml), $FeCl_2$ (1 mM, 5 mM, 10 mM), hemin (0.01 mM, 0.1 mM, 1 mM), haemoglobin (1 mg/ml, 10 mg/ml, 50 mg/ml), holo-transferrin (10 mg/ml, 25 mg/ml, 50 mg/ml), and lactoferrin (10 mg/ml, 25 mg/ml, 50 mg/ml) were spotted directly onto the plates and were incubated 48 to 72 h at 37 °C. Plates without iron sources added were used as negative control. The growth of bacteria was detected by visual inspection. All iron sources were purchased from Macklin, China.

## Biofilm formation assays

Biofilm formation assays were performed on the strains above following a previously described method[31]. For each strain, bacterial cells harvested from overnight culture were resuspended in PBS to a turbidity equivalent to that of a 0.5 McFarland standard and further inoculated 1:100 into fresh LB medium. Aliquots (200 µl, $n = 12$) of the inoculated LB were added into a 96-well polystyrene microplate and were incubated at 37 °C for 24 h. The microplate was then washed three times with PBS and air-dried for 30 mins. Biofilms in the wells were stained with 250 µl 1% (w/v) crystal violet for 20 mins at room temperature. The stained wells were again washed with distilled water to remove unbound stain and allowed to dry for 30 mins. The stained biofilms were treated with 200 µl 33% acetic acid and $OD_{570}$ was recorded using a fluorometer plate reader (BioTek Epoch2, USA). *Acinetobacter baumannii* ATCC 27853 and fresh LB were used as positive and negative controls. Optical density cutoff (ODc) was calculated as the average OD of the negative control plus three times the standard deviation of the negative control as proposed previously[30,31]. An average OD value ≤ ODc, >ODc and ≤2x ODc, >2x ODc and ≤ 4x ODc, and >4x ODc indicates non-, weak, moderate and strong biofilm production, respectively.

## Statistics and reproducibility

Statistical analyses were performed using GraphPad Prism (version 9.3.1). Level of significance between two groups was assessed with two-tailed unpaired Student's $t$ test. Survival analysis for wax moth larvae infected with different bacterial strains was performed using Log-rank (Mantel−Cox) test with GraphPad Prism (version 9.3.1). All information on sample sizes and statistics can be found in the figure legends and the reporting summary. No statistical method was used to predetermine sample size. No data were excluded from the analyses. Wax moth larvae were randomly allocated into different groups. Other experiments were not randomised. The investigators were not blinded to allocation during experiments and outcome assessment.

## Data visualisation

Bar charts, violin plots and line charts were generated using GraphPad Prism (version 9.3.1). Maps were generated using ggplot2[68] and sf[69] packages. Phylogenetic trees were annotated and visualised with iTOL[54], ggtree v3.2.1[63] or Phandango[70]. Pairwise alignment and visualisation of the genomic sequences were carried out using clinker v0.0.28[58].

## Reporting summary

Further information on research design is available in the Nature Portfolio Reporting Summary linked to this article.

## Data availability

Sequence data and genome assemblies generated in this study have been submitted to GenBank under the BioProject PRJNA931432 and PRJNA951454. The individual Illumina sequence read accession numbers of ST410 isolates are listed in Supplementary Data 2. The accession numbers for the nanopore sequenced isolates in this study can be found in Table S1. Source data are provided with this paper.

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

## Acknowledgements

This work was undertaken as part of the DETECTIVE research project funded by the National Natural Science Foundation of China (8181101332) and the Medical Research Council (MR/S013660/1). W.v.S. was also supported by a Wolfson Research Merit Award (WM160092). We thank Dr. Andries J. van Tonder for guidance on the bioinformatics analysis.

## Author contributions

C.Z., M.A.H., A.M., W.v.S., and X.B. initiated and designed the study. X.B., Y.G., R.A.M., and E.L.D. performed the experiments and analyses. B.L. and L.Y. performed experiments. Y.G., B.L., J.L., N.H., S.S., and Y.L. collected and provided the bacterial isolates. X.B. worte the manuscript with input from C.Z., M.A.H, A.M., W.v.S., Y.G., and R.A.M. All the authors reviewed the manuscript.

## Competing interests

The authors declare no competing interests.
