## [Peer Review File · Nature Communications]

REVIEWER COMMENTS

Reviewer #1 (Remarks to the Author):

In this article, the authors detail analysis of 388 carbapenem-resistant *Escherichia coli* from Chinese hospitals collected between 2017 and 2021. The main finding was that ST410 isolates predominated ($n = 109$) with many of these isolates coming from a childrens' hospital which were then further analyzed. They obtained public sequences of a total of nearly 1,000 ST410 isolates and grouped these strains with a previously reported multi-drug resistant clone known as B4/H24RxC (PMID: 31482141 and others although naming is somewhat inconsistent across studies). The B4/H24RxC clone has been described by several groups as generally containing OXA-181 (occasionally NDM-5) on an IncX3 plasmid. In figure 3, the authors show that their strains (and a smattering of others) cluster distinctly from the B4/H24RxC strains so they call them B5/H24RxC. These strains tend to contain NDM-5 on an F-type plasmid that has been previously present in the B4 lineage but did not contain NDM-5 (shown in Figure 5). The B5 strains differed from the B4 strains by many changes including a high pathogenicity island (11 genes) encoding the siderophore yersiniabactin. The authors estimated the emergence of B5 at about 2008. Using publicly available sequences, they found 174 B5 strain, 155 of which were in China or Southeast Asia and nearly all the remaining from Australia. There were a few strains (maybe 2 or 3?) identified from the US and one or so each from Denmark, Switzerland, and Nigeria. The B5 strains grew better, killed moth larva faster, and used iron better than B4 strains.

Overall, the authors are advancing knowledge regarding ST410 carbapenem resistant *E. coli*. The main noteworthy finding is the thorough description of a new clone of carbapenem resistant ST410 that may have increased clinical impact relative to its precursors due to acquisition of a pathogenicity island. Although there have been several papers in this area, the authors do an excellent job with a detailed analysis of both chromosomal and plasmid elements present in this clone as well as comparing them to its precursor.

Carbapenem resistant ST410 is clearly a major concern and thus the paper should be of general interest. Several papers have been published on this subject with similar phylogenomics etc (for example many papers show downloads of available ST410 genomes). including one that includes a co-author of this paper entitled, "Key evolutionary events in the emergence of a globally disseminated, carbapenem resistant clone in the *Escherichia coli* ST410 lineage" which described the B4 clone (and repeats much of the data herein such as the molecular clocks, iron growth etc.). Thus, the novelty of the work is somewhat blunted.

The methods are sound with sufficient detail provided to reproduce the work. The conclusions are generally appropriate although calling the clone "globally disseminated" is likely an over-reach at present based on a few genomes being publicly available from distinct locales. The F type plasmid

carrying NDM-5 that the authors describe herein seems to be the same as the F type, NDM-carrying plasmid of “clade B3” described in Chen et al, PMID: 36204966.

Specific suggestions:

1. Please clarify relationship between F type plasmid from this study with that of Chen et al.
2. Line 90 and elsewhere – isolating CRE from urine or sputum does not mean infection so would modify language given that no clinical data is presented.
3. Line 103 and elsewhere – it is not clear to me that this a true “Chinese epidemiologic surveillance study” as opposed to sampling a limited number of hospitals and so it is problematic to compare these data to other studies and claim changing trends
4. Figure 2C – the large circles obscure the tree branches. Would shrink to make tree more visible.
5. Figure 3 – the key conclusion from this tree (and indeed from this entire study) is that B5 is a separate clade from B4. Please provide the bootstrap values that support this conclusion.
6. Figure 4 – it not clear how the number of CREC per country (the circles) was determined in panel A. Also panel A does not really add anything over panels B and C. Moreover, concluding that a clone is “globally disseminated” because one or two isolates are found on a different continent (assuming that someone did not put in wrong metadata) is a stretch. Moreover, perhaps someone from China or Southeast Asia visited those countries and the isolate came from them with no further spread. Nearly all non-China/Southeast Asia B5 strains came from Australia which is not surprising given the geographic proximity.
7. Figure 8a - 2 hrs (line 255) is too long as a doubling time for E. coli (typical is 20-30 mins). Doubling times should be measured based on the exponential growth period.
8. Figure 8b- doing a competition assay by measuring AMR genes which are present on mobile genetic elements is problematic. These genes are known to amplify and otherwise have unstable copy numbers so should not be used to track relative organism burden.
9. Given that the previous B4 ST410 paper (PMID: 31482141) showed that the B4 strains grew well with Lactoferrin, the authors should discuss why they seemed to get discordant results (no growth for either the B4 or B5 strains).

Reviewer #2 (Remarks to the Author):

Ethics

I could find no mention of ethics for the collection of CREC isolates reported inn this study. A statement about what permissions were required to collect these data should be included in the manuscript,

particularly as supp table 3 reports the age, sex, admission, and length of stay for the patients from which the CREC isolates were included in this study.

Introduction

The introduction was overall well-written. However, the authors make several assumptions about readership level of knowledge of CREC including specific nomenclature and AMR beyond carbapenems.

- Significance of the nomenclature of A/H53 and B/H24 (and then sub-lineages within this group). This is important as B4/H24RxC and B5/H24RxC are used extensively in the manuscript and what this nomenclature signifies is not explained. Think links to the fim-typing (reported in methods)? In the discussion the authors note the classification scheme (L297-299) – this should be included and explained in the introduction.
- Authors state these are MDR lineages with emerging resistance to carbapenems – unclear what the definition is that the authors are using for MDR (>3 drug classes) and if there is a specific AMR profile associated with these clones (beyond the carbapenem resistance).
- Authors mention PBP3 but do not explain what this is or the significance of this to the reader. It is then not referred to in the methods, results or discussion so perhaps is not needed?

Methods

- Five isolates were selected for ONT – unclear what the rationale was for these ones. Also assume that the five are those given in supp fig 3? There is also a B4/H24 isolate in this figure – was this also ONT for this study? Could only see five of these as complete genomes in the bioproject specified. Also couldn't find any details of the accessions for the plasmids reported in Supp Fig 3 – these details should be included either in text /figures or in additional supplementary file so that these plasmid sequences may be readily accessed by other researchers.
- The authors used abricate with several databases to screen for genome content including AMR genes, plasmids etc. What parameters were used with abricate? Default or something else? These are important details. Further, the authors need to cite the respective databases used in conjunction with abricate and provide the version of each database. Finally, the authors may wish to consider using the recently published abritAMR tool that is ISO-accredited for AMR (Sherry et al 2023 DOI <https://doi.org/10.1038/s41467-022-35713-4>) and the ECTyper tool (Besanov et al 2021 <https://doi.org/10.1099/mgen.0.000728>) for OH which was benchmarked as best performing OH in silico typing for E. coli
- The authors mention the Ogn5 in the results but don't describe these comparative analyses in the methods. Please address.
- Treemmer – were any other parameters considered in subsampling the larger ML tree to 500 isolates. E.g year of collection or geographic location in addition to retaining the diversity?

- The authors undertook a lot of work trying different models in BEAST. How did the authors assess that the best fit model was HKY with a strict clock and Bayesian skyline demographic prior?
- There were some differences in the types of alignments being generated (sections Phylogenetic analysis of *E. coli* ST410 and Comparative genomic analysis of genomics of the B4/H24RxC and B5/H24RxC). E.g phaster was used for the second section but not used to identify and exclude phage regions in the overall alignment. Would these differences impact some of the results, e.g. calculating the pair-wise SNP distances?
- How were the isolates selected for the phenotypic experiments. For growth curves random selection apart from blaOXA-181 + B4/H24RxC and blaNDM-5 + B5/H24RxC. And virulence assays etc – why were the 2 B4/H24RxC and 6 B5/H24RxC isolates selected? Were all isolates characterised for biofilm formation?

Results

- Carbapenem resistance. The authors report the median MIC values for ertapenem (L95). In Supp Table 1 >200 isolates were tested as R by disc diffusion. How were these data treated when determining the median and also visualising the data in the violin plots? Further, there was no mention of disc diffusion in the methods – please address. Was the phenotype and genotype data concordant for carbapenem resistance? This relates to the methods point raised re abricate parameters.
- Four groups were identified in the ST410 isolates from the children’s hospital L116-L132. It was unclear what threshold was applied to delineate the between the groups based off pairwise SNP distances and why this threshold was selected (different to pairwise SNP distances within groups). From figure 2C – looks like two groups with two singletons? The author may wish to consider referring to Gorrie et al 2021 ([https://doi.org/10.1016/S2666-5247\(21\)00149-X](https://doi.org/10.1016/S2666-5247(21)00149-X)) for parameters to transmission thresholds in hospital settings.
- BAPS of global ML tree. The authors classified the ST410 isolates into groups with BAPS and reported to level 1. What was the second level of BAPS clustering and did this correlate to the identified B4/H24RxC and B5/H24RxC groups? The authors should already have these data as think BAPs reports to the second level under default parameters. This would provide additional support for the groups identified. How did the BAPS clusters correlate to the SNP-distance based threshold groups? The authors state on L142 that all isolates from this study fell into BAP1 but in Fig 3A – there are multiple isolates in BAP2 lineage that have the red star indicative that these isolates were from this study as well. From Fig 3, it appears as though these isolates also have blaNDM-5 (the green ring) but lack the IncF plasmid replicon? It was unclear what this meant.
- AMR screening. The authors screened for all genes in the ResFinder database but only reported a limited number of AMR genes. In some figures, genes mediating resistance to e.g. co-trimoxazole are named (Fig 5B) or total number of ARGs (Fig 3B) and in L100-101 eleven unspecified isolates are noted as carrying a mcr-1 gene but the overall presence/ absence of AMR genes isn’t reported. This could easily be a supp table/ supp figure and is useful background information for the reader. It would also help to address the point of MDR raised above in the intro. Further, in Fig6B, it appears as though

blaCMY-2 (ESBL associated) is chromosomally integrated in the isolates shown. Is this gene present in all ST410? Could find no other mention of it in the manuscript or supplementary materials.

- The difference in virulence from the wax larvae model was shown between the B4 and B5 groups. Did the B5 isolates selected for testing have the HPI (reported in L156 as detected in 77.6% (135/174). Could there have been another reason apart from HPI for survival rate? e.g the authors note the difference in O-antigen and other differences in the accessory genome content.

Discussion

- The authors call the new lineage 'hypervirulent' in the title but it was unclear as to why this lineage is hypervirulent/ what makes something hypervirulent vs virulent? The authors may wish to expand on this more as the authors state that the HPI element is frequently found in ExPEC and enables the bacteria to better colonise and persist (L302-306). It was unclear if the authors were also suggesting the HPI also mediated enhanced virulence? Or if the reason for hypervirulence was not yet determined?

- The authors discuss differences in the presence of non-synonymous SNPs associated with B5/H24RxC (L317-325) and link through to some potential phenotypic changes. Unclear if this is accurate given not testing any of the SNPs explicitly. E.g. the significance of the biofilm formation and YeeJ discussion (L322-325) was unclear as the results (L272 -L275) stated both populations were poor biofilm performers and as such, the difference in YeeJ doesn't appear to be important.

- The authors suggest geographical spread from south-east Asia to other countries (L331-L333). While this is likely, it is a generalised statement that requires additional references (e.g from other studies demonstrating the emergence and dissemination of CREC / MDR E. coli in the region) or additional work from the authors e.g. phylogeographical analyses/ ancestral state reconstruction to support this statement.

- The strengths, limitations and future directions of this study were not really addressed in the discussion. A note was made of sampling bias (lack of data from companion animals and food samples) but the authors could consider these aspects of the study in greater detail.

-

Figures + Tables

- There were 8 main figures included in the study + additional supplementary ones. Some of the main ones could be moved to supp e.g. Fig 4 – this was not a key finding to the study but more of a summary of the global dataset. Further, one of the key findings of the study was the novel O-antigen (novel1/OgN5) in the specific group. The comparison was shown in Supp Fig 2B but the authors could consider moving this panel to the main body of text in Fig 3 where they report the differences in O-antigens.

- For all violin plots – please also plot the underlying data

- Fig 6B – the authors have a schematic showing what looks to be chromosomal integration of a blaCMY-2 gene. I could find no mention of the blaCMY-2 gene anywhere in the manuscript, supplementary figures or data tables.

- Fig 2C : GrapeTree visualisation. What does the scale bar indicate? Why is 20_20 highlighted in red (this isolate is not referred to in the main manuscript but can see it was one of the isolates subject to ONT based off Supp Fig 3)?
- Supp Fig 1 – why showing *fyuA*, *fstI* insertions? Refer to *fstI* in Intro (L73) and methods but not in results/ discussion, and *fyuA* as one of the genes on the HPI – using it as a marker for the HPI?
- Supp table 5 sheet 3 – this looks like a duplication of the data on sheet 1?

General

- Number of times use acronyms without expanding either at first use or at all. E.g ARG first used at L151 and explained on L391. CFU, first used at L501, not explained at all.

REVIEWER COMMENTS

Reviewer #1 (Remarks to the Author):

In this article, the authors detail analysis of 388 carbapenem-resistant *Escherichia coli* from Chinese hospitals collected between 2017 and 2021. The main finding was that ST410 isolates predominated (n = 109) with many of these isolates coming from a childrens' hospital which were then further analyzed. They obtained public sequences of a total of nearly 1,000 ST410 isolates and grouped these strains with a previously reported multi-drug resistant clone known as B4/H24RxC (PMID: 31482141 and others although naming is somewhat inconsistent across studies). The B4/H24RxC clone has been described by several groups as generally containing OXA-181 (occasionally NDM-5) on an IncX3 plasmid. In figure 3, the authors show that their strains (and a smattering of others) cluster distinctly from the B4/H24RxC strains so they call them B5/H24RxC. These strains tend to contain NDM-5 on an F-type plasmid that has been previously present in the B4 lineage but did not contain NDM-5 (shown in Figure 5). The B5 strains differed from the B4 strains by many changes including a high pathogenicity island (11 genes) encoding the siderophore yersiniabactin. The authors estimated the emergence of B5 at about 2008. Using publicly available sequences, they found 174 B5 strain, 155 of which were in China or Southeast Asia and nearly all the remaining from Australia. There were a few strains (maybe 2 or 3?) identified from the US and one or so each from Denmark, Switzerland, and Nigeria. The B5 strains grew better, killed moth larva faster, and used iron better than B4 strains.

Overall, the authors are advancing knowledge regarding ST410 carbapenem resistant *E. coli*. The main noteworthy finding is the thorough description of a new clone of carbapenem resistant ST410 that may have increased clinical impact relative to its precursors due to acquisition of a pathogenicity island. Although there have been several papers in this area, the authors do an excellent job with a detailed analysis of both chromosomal and plasmid elements present in this clone as well as comparing them to its precursor.

Carbapenem resistant ST410 is clearly a major concern and thus the paper should be of general interest. Several papers have been published on this subject with similar phylogenomics etc (for example many papers show downloads of available ST410 genomes). including one that includes a co-author of this paper entitled, "Key evolutionary events in the emergence of a globally disseminated, carbapenem resistant clone in the *Escherichia coli* ST410 lineage" which described the B4 clone (and repeats much of the data herein such as the molecular clocks, iron growth etc.). Thus, the novelty of the work is somewhat blunted.

The methods are sound with sufficient detail provided to reproduce the work. The conclusions are generally appropriate although calling the clone "globally disseminated" is likely an over-reach at present based on a few genomes being publicly available from distinct locales. The F type plasmid carrying NDM-5 that the authors describe herein seems to be the same as the F type, NDM-carrying plasmid of "clade B3" described in Chen et al, PMID: 36204966.

Specific suggestions:

1. Please clarify relationship between F type plasmid from this study with that of Chen et al.

We thank the reviewer for their comment. We can confirm that the F type plasmid (FII-1:FIA-1:FIB-49) in this study is the same as that was described by Chen et al. We have added a couple of sentences to clarify in the manuscript. Please see Line 336-341.

2. Line 90 and elsewhere – isolating CRE from urine or sputum does not mean infection so would modify language given that no clinical data is presented.

The language has been modified by changing "most associated..." to "possible association" in order to give a more accurate description. Please see Line 96-97.

3. Line 103 and elsewhere – it is not clear to me that this a true “Chinese epidemiologic surveillance study” as opposed to sampling a limited number of hospitals and so it is problematic to compare these data to other studies and claim changing trends

We thank the reviewer for raising this point. We’ve modified the manuscript by changing “epidemiology” to “population” (or deleting the word) to better describe this study. We used a commonly employed study design which captures a clinically relevant population of bacteria by collecting CREC isolates from hospitals across 26 Chinese provinces between 2017 and 2021. It included 26 out of 31 provinces in mainland China, covering a vast area of the country. Although it does have a relatively narrow sampling frame (5 years), it provides valuable comparative data in addition to providing primary data about bacterial populations.

We compared the data with two previously published studies on CREC prevalence in China (PMID: 34535075, PMID: 28479289) which used a very similar approach by screening carbapenem-resistant Enterobacteriaceae (including CREC) in Chinese hospital from different provinces. Therefore, we consider it is of value to compare our finding to these previously published data.

In study PMID 34535075, the most isolated CREC belonged to ST167, followed by ST410 and ST131. And in study PMID 28479289, the most isolated CREC belonged to ST131, followed by ST167 and ST410. In this study, ST410 was the most commonly isolated CREC ST, suggesting a shift in lineage prevalence.

4. Figure 2C – the large circles obscure the tree branches. Would shrink to make tree more visible.

To display the tree branches clearer, we now present the phylogeny in Figure 2c in a circular format instead of the figure generated using GrapeTree. Description for Figure 2c in the figure legend has been changed accordingly. Line 818-820.

5. Figure 3 – the key conclusion from this tree (and indeed from this entire study) is that B5 is a separate clade from B4. Please provide the bootstrap values that support this conclusion.

We have modified Figure 3a to include the bootstrap values in the phylogeny. The bootstrap value for the branch separating B5/H24RxC from B4/H24RxC is 100. Description for Figure 3a in the figure legend has been changed accordingly. Line 825.

6. Figure 4 – it not clear how the number of CREC per country (the circles) was determined in panel A. Also panel A does not really add anything over panels B and C. Moreover, concluding that a clone is “globally disseminated” because one or two isolates are found on a different continent (assuming that someone did not put in wrong metadata) is a stretch. Moreover, perhaps someone from China or Southeast Asia visited those countries and the isolate came from them with no further spread. Nearly all non-China/Southeast Asia B5 strains came from Australia which is not surprising given the geographic proximity.

We thank the reviewer for pointing out this mistake. Circles in panel A in fact represent the total numbers of *E. coli* ST410 (not CREC) in each country and the inverted triangles represent the presence of the B5/H24RxC clone within the country. We have corrected this mistake in the figure and changed figure description accordingly. Also, taking reviewer 2’s comment into consideration, we have moved Fig 4 to supplementary figures (now Fig. S1).

We agree that the reviewer may have a valid point that it may be a stretch to call the B5/H24RxC “globally disseminated”, given the smaller numbers of isolates found in some continents or countries in our analysis. Therefore, we decided to screen recently available *E. coli* ST410 genomes for more B5/H24RxC and examine whether it is proper to call this clone “globally disseminated”.

To capture a more up-to-date picture of the spread of the B5/H24RxC clone worldwide since our analysis, we screened a new collection of *E. coli* ST410 genomes (n=714, Supplementary Data Set 8) available from 14 Jan 2022 to 27 Oct 2023 from Enterobase. A further 84 B5/H24RxC genomes were identified from the collection (Fig. S10, Supplementary Data Set 8). The isolates providing these genomes were collected between 2016 and 2023 across 13 countries on 5 continents (Fig. S10), mainly from humans, with 2 isolates from dogs and 2 from environmental samples. The countries with the most isolates were Thailand (n=26), United States (n=17), Germany (n=7), China (n=6), Cambodia (n=6), Australia (n=5) and Norway (n=4), suggesting the ongoing spread of this clone globally.

This new analysis identified 6 additional countries with the presence of B5/H24RxC clone, bringing the total number of the countries to 17. It is worth noting that the numbers of the isolates have increased in all continents. Taking both analysis into account, we believe this clone is indeed “globally disseminated”. Figure below is Fig S10 which represents the results of the new analysis.

Fig. S10: Analysis on the recent collection of ST410 and B5/H24RxC clone. (a) Midpoint rooted maximum-likelihood phylogeny of 714 newly available ST410 genomes from Enterobase and 388 genomes (214 B4/H24RxC and 174 B5/H24RxC) from the original analysis, constructed using a core genome SNP alignment generated by Snippy v4.6.0 with strain 020026 (Genbank accessions CP034954 to CP034958) as the reference. Branch support was performed with 1,000 bootstrap replicates. (b) An enlarged phylogenetic tree showing the B4/H24RxC and B5/H24RxC clones with selected genomic characteristics. (c) Global distribution of the newly identified B5/H24RxC isolates in the recent ST410 collection. Total number of B5/H24RxC isolates in each country is within the blue circles.

7. Figure 8a - 2 hrs (line 255) is too long as a doubling time for *E. coli* (typical is 20-30 mins). Doubling times should be measured based on the exponential growth period.

We agree with the reviewer’s comment that the doubling time appears too long for *E. coli*. After re-examining the data, we included the growth data generated using full strength LB,

half strength LB and one-tenth strength LB and used a previously published Python script available at https://github.com/huoww07/calulate_bacteria_doubling_time to calculate the doubling time of the isolates.

However, the doubling time calculated is still considerably longer than 20-30 mins. It most likely was affected by the reduced agitation during shaking in the optical growth analyser and the smaller volume compared with a larger liquid culture as suggested previously [1,2] The table below presents the doubling time (mean \pm SD) of the selected isolates in different strength of LB. It shows that in $\frac{1}{2}$ strength and $\frac{1}{10}$ strength LB, B5/H24RxC isolates grew significantly faster than B4/H24RxC isolates ($p < 0.0001$). In full strength LB, some B5/H24RxC isolates (18-13, 19-3 and 19-17) grew faster than B4/H24RxC isolates while some B5/H24RxC isolates (20-11, 20-25 and 20-30) had a similar doubling time with B4/H24RxC isolates and the control strain. We suspect that full strength LB may be too nutrient rich and camouflaged the difference in growth rates of the two clones. In the manuscript, we therefore presented the growth data in $\frac{1}{2}$ LB in Figure 7a and 7b. Growth data in full strength LB and $\frac{1}{10}$ LB were presented in Figure S8.

Table S2: Doubling time (mins) of selected isolates in different strength of LB

	B5/H24RxC					
	18-13	19-3	19-17	20-11	20-25	20-30
LB	52.591 \pm 0.146	55.032 \pm 0.433	61.527 \pm 1.241	82.284 \pm 1.307	85.124 \pm 0.109	83.348 \pm 2.627
1/2 LB	63.241 \pm 0.733	60.591 \pm 1.007	71.076 \pm 0.995	70.213 \pm 0.083	73.195 \pm 0.647	72.596 \pm 0.154
1/10 LB	58.763 \pm 26.031	98.218 \pm 2.264	116.800 \pm 1.918	137.436 \pm 2.601	158.813 \pm 3.176	134.506 \pm 2.511

	B4/H24RxC		Control
	005828	045869	ATCC 25922
LB	84.447 \pm 0.593	83.463 \pm 1.361	90.643 \pm 0.531
1/2 LB	82.715 \pm 1.251	84.354 \pm 2.012	86.875 \pm 2.074
1/10 LB	211.193 \pm 17.856	212.242 \pm 18.882	535.183 \pm 117.962

1. Hecht, A., Filliben, J., Munro, S.A. et al. A minimum information standard for reproducing bench-scale bacterial cell growth and productivity. *Commun Biol* 1, 219 (2018). <https://doi.org/10.1038/s42003-018-0220-6>
2. Running, J. A. & Bansal, K. Oxygen transfer rates in shaken culture vessels from Fernbach flasks to microtiter plates. *Biotechnol. Bioeng.* 113, 1729–1735 (2016).

8. Figure 8b- doing a competition assay by measuring AMR genes which are present on mobile genetic elements is problematic. These genes are known to amplify and otherwise have unstable copy numbers so should not be used to track relative organism burden.

We have repeated the competition assay by measuring the relative expression of chromosomal genes *fyuA* and *yodB* in the same culture.

Genomic analysis revealed that these isolates of the B5/H24RxC clone possessed gene *fyuA* but not *yodB*, while isolates of the B4/H24RxC clone contained gene *yodB* but not *fyuA*. The new qPCR results agreed with the conclusion we had come to using the AMR genes and confirmed that isolates of B5/H24RxC grew faster than isolates of B4/H24RxC, as demonstrated by the increasing ratio of *fyuA* to *yodB* levels in the same culture. This result also agrees with the conclusion based on the growth curve analysis. Please see changes made in Line 272-277 and 547-562 and Fig 7a-d.

9. Given that the previous B4 ST410 paper (PMID: 31482141) showed that the B4 strains grew well with Lactoferrin, the authors should discuss why they seemed to get discordant results (no growth for either the B4 or B5 strains).

We were also surprised to see this discordance with the previous paper (PMID: 31482141). However, repeats of the assay gave the same results. It is possible that we used different strains from the ones used in the previous paper, and these strains happened to behave differently for unknown reasons. However, it may suggest that not all B4/H24RxC isolates could utilise iron from lactoferrin.

We have added a sentence in the manuscript to highlight this difference to readers. Line 294-298.

Reviewer #2 (Remarks to the Author):

Ethics

I could find no mention of ethics for the collection of CREC isolates reported in this study. A statement about what permissions were required to collect these data should be included in the manuscript, particularly as supp table 3 reports the age, sex, admission, and length of stay for the patients from which the CREC isolates were included in this study.

We thank the reviewer for this important comment. We have added an "Ethics and consent" section in the manuscript. Line 402-407.

"This study was approved by the medical ethics committee of the First Affiliated Hospital of Guangzhou Medical University on 21 May, 2018. For the work involving the CREC isolates from the children's hospital, ethics was approved by the medical ethics committee of the Children's Hospital of Soochow University on 5 January, 2021. Individual consent was obtained from the patients' guardians by hospital staff."

We removed the "date of birth" and "age in days" columns in the Supplementary Data 3, and replacing these with an "Age group (days)" column to improve anonymity in the reporting.

Introduction

The introduction was overall well-written. However, the authors make several assumptions about readership level of knowledge of CREC including specific nomenclature and AMR beyond carbapenems.

- Significance of the nomenclature of A/H53 and B/H24 (and then sub-lineages within this group). This is important as B4/H24RxC and B5/H24RxC are used extensively in the manuscript and what this nomenclature signifies is not explained. Think links to the fim-typing (reported in methods)? In the discussion the authors note the classification scheme (L297-299) – this should be included and explained in the introduction.

We have added more information in the manuscript to properly introduce A/H53 and B/H24 and the sub-lineages in B/H24. Please see Line 63-69.

- Authors state these are MDR lineages with emerging resistance to carbapenems – unclear what the definition is that the authors are using for MDR (>3 drug classes) and if there is a specific AMR profile associated with these clones (beyond the carbapenem resistance).

We apologise for not stating the definition of multidrug resistance (MDR). We have now added a sentence in the introduction to introduce the definition. Please see Line 40-41.

We have also rewritten some sentences to emphasise the association between specific AMR genes and the B/H24 sub-lineage. Please see line 63-69.

- Authors mention PBP3 but do not explain what this is or the significance of this to the reader. It is then not referred to in the methods, results or discussion so perhaps is not needed?

PBP3 (penicillin-binding protein 3, also termed FtsI) is encoded by the *ftsI* gene. Some PBP3 amino acid insertions (i.e. YRIN, YRIK or TIPY) are reported to associated with susceptibilities to several β -lactams, including aztreonam, ceftazidime, cefepime, ceftazidime/avibactam and ceftolozane/tazobactam [1,2]. In the Chen *et al.* Paper [3], the authors characterised ST410-

B3 by the acquisition of a four-amino-acid (YRIN) insertion in the *ftsI*-encoded. We would argue this is important information for understanding the proposed classification scheme for B/H24 proposed by Chen *et al.* and it is better to include the information in the manuscript. A couple of sentences were added to provide details about PBP3/FtsI. Please see Line 73-78.

We did indeed include the method for PBP3 insertion identification, although we have used FtsI instead of PBP3. We have put "(PBP3)" following FtsI to avoid any confusion (Line 461). The result for FtsI (PBP3) insertion analysis is already included in Supplementary Fig. S2.

[1]. Alm RA, Johnstone MR, Lahiri SD. Characterization of Escherichia coli NDM isolates with decreased susceptibility to aztreonam/avibactam: role of a novel insertion in PBP3. *J Antimicrob Chemother* 2015; **70**(5): 1420-8.

[2]. Zhang Y, Kashikar A, Brown CA, Denys G, Bush K. Unusual Escherichia coli PBP 3 Insertion Sequence Identified from a Collection of Carbapenem-Resistant Enterobacteriaceae Tested In Vitro with a Combination of Ceftazidime-, Ceftaroline-, or Aztreonam-Avibactam. *Antimicrob Agents Chemother* 2017; **61**(8).

[3]. Chen L, Peirano G, Kreiswirth BN, Devinney R, Pitout JDD. Acquisition of genomic elements were pivotal for the success of Escherichia coli ST410. *J Antimicrob Chemother* 2022.

Methods

- Five isolates were selected for ONT – unclear what the rationale was for these ones. Also assume that the five are those given in supp fig 3? There is also a B4/H24 isolate in this figure – was this also ONT for this study? Could only see five of these as complete genomes in the bioproject specified. Also couldn't find any details of the accessions for the plasmids reported in Supp Fig 3 – these details should be included either in text /figures or in additional supplementary file so that these plasmid sequences may be readily accessed by other researchers.

All five isolates belong to B5/H24RxC clone, they were specifically selected based on their genomic differences (i.e. covering/representing the range of that diversity) as suggested by their positions in the B5/H24RxC phylogeny. And their genomic differences were revealed by the long-read nanopore sequencing, as shown in Fig. S3 (updated to Fig. S6).

We apologise for failing to provide a more informative figure legend for Fig. S3 (updated to Fig. S6). The B4/H24RxC isolate 020026 was not nanopore sequenced in this study. It is used in this study as a reference for several analyses. We have now added the accession number for 020026 (Genbank: CP034954 to CP034958) in the figure legend. The accession numbers for the five nanopore sequenced isolates in this study, including the accession numbers for both the chromosomes and plasmids can be found in Table S1. Figure legend for Fig. S3 (updated to Fig. S6) has also been modified accordingly.

Table S1: Genbank accessions for all B5/H24RxC isolates in BioProject PRJNA951454

Isolate	chromosome/plasmids	GenBank	Size (bp)	GC content (%)
E22	chromosome	CP123036.1	4,826,658	50.5
	pE22P1	CP123037.1	108,280	51.5
	pE22P2	CP123038.1	3,376	55
	pE22P3	CP123039.1	2,088	47
18-4	chromosome	CP123013.1	4,847,302	50.5
	p18-4P1	CP123014.1	107,858	51.5
	p18-4P2	CP123015.1	3,376	55
	p18-4P3	CP123016.1	2,088	47
20-16	chromosome	CP123024.1	4,844,449	50.5
	p20-16P1	CP123025.1	100,602	51.5
	p20-16P2	CP123026.1	3,373	55
	p20-16P3	CP123027.1	2,088	47
	p20-16P4	CP123028.1	1,552	51.5

19-7	chromosome	CP123017.1	4,838,085	50.5
	p19-7P1	CP123018.1	93,917	47.5
	p19-7P2	CP123019.1	83,900	50.5
	p19-7P3	CP123020.1	67,953	52.5
	p19-7P4	CP123021.1	3,376	55
	p19-7P5	CP123022.1	2,088	47
	p19-7P6	CP123023.1	1,552	51.5
20-20	chromosome	CP123029.1	4,799,026	50.5
	p20-20P1	CP123030.1	107,778	47
	p20-20P2	CP123031.1	84,132	50.5
	p20-20P3	CP123032.1	39,066	61.5
	p20-20P4	CP123033.1	3,376	55
	p20-20P5	CP123034.1	2,088	47
	p20-20P6	CP123035.1	1,552	51.5

- The authors used abricate with several databases to screen for genome content including AMR genes, plasmids etc. What parameters were used with abricate? Default or something else? These are important details. Further, the authors need to cite the respective databases used in conjunction with abricate and provide the version of each database. Finally, the authors may wish to consider using the recently published abritAMR tool that is ISO-accredited for AMR (Sherry et al 2023 DOI <https://doi.org/10.1038/s41467-022-35713-4>) and the ECTyper tool (Besanov et al 2021 <https://doi.org/10.1099/mgen.0.000728>) for OH which was benchmarked as best performing OH *in silico* typing for *E. coli*

We used ABRicate v1.0.1 with default parameters, which is now specified in the manuscript. The citations for the respective databases (with the date when they were updated) used have been added in the manuscript. Please see the changes made in Line 455-458.

We thank the reviewer for recommending the use of abritAMR which identifies both acquired AMR genes and gene mutations that confer resistance. We have re-analysed the genomes using abritAMR and updated the method and result accordingly in the manuscript. Line 453-455. Figure 3b.

We also thank the reviewer for recommending the use of ECTyper for OH typing in *E. coli*. However, we found ECTyper v1.0.0 with database version 1.0 failed to type the novel O-antigen identified by ABRicate v1.0.1 using EcOH database (updated on 15 September 2022). Therefore, we have not modified the OH *in silico* typing method and results in the manuscript.

- The authors mention the Ogn5 in the results but don't describe these comparative analyses in the methods. Please address.

We thank the reviewer for pointing this out. The method for the comparative analysis for O-antigen Onovel1 and a previously reported Ogn5 has been added into the methods. Please see Line 505-508.

- Treemmer – were any other parameters considered in subsampling the larger ML tree to 500 isolates. E.g year of collection or geographic location in addition to retaining the diversity?

No other parameters were used when subsampling the larger ML tree. Upon careful examination, the subsampled tree (500 genomes) has a very good representation (see the figures below) of the isolation years and countries in comparison to the larger tree (956 genomes). In the larger tree, a total of 29 countries were included while 28 countries were included in the subsampled tree. All collection years apart from 1984 were included in the subsampled tree. In the whole dataset, there was only one isolate collected in 1984 from the United States. We have edited the manuscript to reflect this. See Line 512-513. The following figure is included as Fig. S4.

Fig. S4: (a) Bar chart showing the distribution of all 956 ST410 *E. coli* isolates in countries and in collection years. (b) bar chart showing the distribution of Treemmer retained ST410 *E. coli* isolates (n=500) in countries and in collection years.

- The authors undertook a lot of work trying different models in BEAST. How did the authors assess that the best fit model was HKY with a strict clock and Bayesian skyline demographic prior?

We thank the reviewer for raising this question. We selected the best model by examining the log files from each model after 100 million Markov chain Monte Carlo (MCMC) iterations using Tracer v1.7.2. The model chosen had the best convergence and effective sample size (ESS).

However, we acknowledge there are alternative ways of assessing the models other than simply checking ESS value. We have now performed model selection using the Nested Sampling package within the BEAST2 program and the best model was found to be a GTR model and a relaxed molecular clock with a Bayesian Skyline population model. This model was different from the model we chose before. Therefore, we performed the BEAST analysis using this model which gave a slightly different but very similar result. The differences in estimating of the most recent common ancestor (TMRCA) for *E. coli* ST410 and the new clone B5/H24RxC using both models were not significant. Changes have been made in relevant parts in the manuscript. Line 256-264, 523-529. Fig. 6.

- There were some differences in the types of alignments being generated (sections Phylogenetic analysis of E. coli ST410 and Comparative genomic analysis of genomics of the B4/H24RxC and B5/H24RxC). E.g phaster was used for the second section but not used to identify and exclude phage regions in the overall alignment. Would these differences impact some of the results, e.g. calculating the pair-wise SNP distances?

For accuracy, we have used the same method to generate whole genome alignment in both sections. The global ST410 phylogeny (n=956) and the phylogeny containing only B4/H24RxC and B5/H24RxC isolates (n=389) were generated using the same method. Briefly, we used Snippy v4.6.0 to produce a full-length whole genome alignment. The full-length whole genome alignment was cleaned with the snippy-clean function and then used as an input to Gubbins v2.4.11 for identifying and filtering regions of homologous recombination. Variant sites in the alignment were extracted using SNP-sites v2.5.12 and a maximum-likelihood (ML) phylogenetic tree was reconstructed using IQ-tree3 accounting for constant sites in the alignment and run for 1000 bootstraps using the extended model selection function. Pairwise SNP distances between isolate genomes were calculated using snp-dists v0.8.2 using the recombination removed alignment from Gubbins as input. Since there is no difference in generating the alignment, we believe that the results would not be affected.

In the above process, PHASTER was not used to identify prophage regions as it was suggested that masking the prophage regions had little to no effect on the core alignment, the core SNP alignment, or pairwise SNP distance [1].

However, PHASTER was used to identify prophage regions in complete genomes (nanopore sequenced and reference strain). Sequence locations of the prophages in these genomes was used to help visualise of the prophage regions in figures. We apologise for not being clear and we have modified the section in the manuscript. Line 498-499.

1. Gorrie CL, Da Silva AG, Ingle DJ, et al. Key parameters for genomics-based real-time detection and tracking of multidrug-resistant bacteria: a systematic analysis. *Lancet Microbe* 2021; 2(11): e575-e83.

- How were the isolates selected for the phenotypic experiments. For growth curves random selection apart from blaOXA-181 + B4/H24RxC and blaNDM-5 + B5/H24RxC. And virulence assays etc – why were the 2 B4/H24RxC and 6 B5/H24RxC isolates selected? Were all isolates characterised for biofilm formation?

The isolates were selected based on their availability and their genetic characteristics. Among the isolates reported in this study, there were only 2 B4/H24RxC isolates that were bla_{OXA-181} positive and bla_{NDM-5} negative and available to us in the lab, and therefore they were initially chosen for the growth competition assay. As for the rationale of selecting the B5/H24RxC isolates, we chose 3 isolates each from Group-1 and Group-2 clones collected from the children's hospital, representing diversity with the B5/H24RxC clone. To maintain consistency, we continued to use these isolates for other phenotypic assays, including virulence assay, iron source growth assay and biofilm formation assay.

We regret to say that we were not able to characterise biofilm formation in all isolates. However, this is justified by the fact that there was no significant difference (in any of the isolates of the two clones tested) from their ability to form biofilm under the assay conditions (both weak formers), which minimises the risk from failing to assay all isolates.

Results

- Carbapenem resistance. The authors report the median MIC values for ertapenem (L95). In Supp Table 1 >200 isolates were tested as R by disc diffusion. How were these data treated when determining the median and also visualising the data in the violin plots? Further, there was no mention of disc diffusion in the methods – please address. Was the phenotype and

genotype data concordant for carbapenem resistance? This relates to the methods point raised re abricate parameters.

There are 220 isolates lacking MIC data for ertapenem. These isolates were all from the China Antimicrobial Surveillance Network (CHINET), and CHINET does not routinely determine ertapenem MIC for CREC, but does test for susceptibility to ertapenem by disc diffusion. The part regarding ertapenem MIC in the manuscript and in Fig 1f were based on the MIC data from the remaining isolates (n=168) in the collection. We apologise for not making this clear. We have now modified the manuscript (Line 101-104) and figure legend (Line 809-910) for Fig 1f to clarify this.

All 388 isolates collected and reported in this study were classified as CREC as they were resistant to at least one of the three carbapenem antibiotics tested (imipenem, meropenem and ertapenem). Following the reviewer's advice, we have re-analysed the AMR profile of the isolates using the recently published abritAMR tool. The predicted carbapenem resistance genes by ABRicate and abritAMR are nearly identical. However, phenotype and genotype data are not concordant for carbapenem resistance as only 87.1% (338/388) of the isolates possessed at least one carbapenem resistance gene. We have updated the manuscript to reflect this (Line 109-111) We have also updated Supplementary Data Set 1 to include the ARGs and gene mutations that confer resistance in these Chinese isolates reported in this study.

- Four groups were identified in the ST410 isolates from the children's hospital L116-L132. It was unclear what threshold was applied to delineate the between the groups based off pairwise SNP distances and why this threshold was selected (different to pairwise SNP distances within groups). From figure 2C – looks like two groups with two singletons? The author may wish to consider referring to Gorrie et al 2021 ([https://doi.org/10.1016/S2666-5247\(21\)00149-X](https://doi.org/10.1016/S2666-5247(21)00149-X)) for parameters to transmission thresholds in hospital settings.

We apologize for not clearly stating the threshold used to infer a transmission event / outbreak. A threshold of ≤ 25 SNPs was applied as recommended by previous studies. Based on this threshold, we identified two possible outbreaks in the children's hospital caused by the isolates in Group-1 and Group-2 as described in the manuscript. Singletons 18-4 and 18-10 did not belong to either groups and each formed a group of their own, but 18-4 was closely related with the isolates in the two main groups as indicated by a pairwise SNP distance of 29 to 48. However, isolate 18-10 was distant from all other isolates with a pairwise SNP distance of 215 to 240. The group classification was supported by a Fastbaps analysis of the isolates, although isolate 18-4 was separated from Group-1 at the second level.

We have edited sentences in the manuscript to clarify this. Please see line 130-143.

- BAPS of global ML tree. The authors classified the ST410 isolates into groups with BAPS and reported to level 1. What was the second level of BAPS clustering and did this correlate to the identified B4/H24RxC and B5/H24RxC groups? The authors should already have these data as think BAPS reports to the second level under default parameters. This would provide additional support for the groups identified. How did the BAPS clusters correlate to the SNP-distance based threshold groups? The authors state on L142 that all isolates from this study fell into BAP1 but in Fig 3A – there are multiple isolates in BAP2 lineage that have the red star indicative that these isolates were from this study as well. From Fig 3, it appears as though these isolates also have blaNDM-5 (the green ring) but lack the IncF plasmid replicon? It was unclear what this meant.

We do indeed have the data on the second level of BAPS clustering. However, B4/H24RxC and B5/H24RxC were not classified into different groups. Given that BAPS was performed using Gubbins-filtered polymorphic sites generated by Snippy, this may be explained by the fact that most (171/204) of the SNPs on the branch separating the B5/H24RxC MDR clone from its most closely related isolate were within the recombination regions (Fig. S5, S7). And

the SNPs outside of the recombination regions were not enough to separate these two groups. This is supported by a new Fastbaps analysis of the two clones using unfiltered core alignment, which classified B4/H24RxC and B5/H24RxC into multiple groups. Figure S3a, S3b.

Similarly, ST410 isolates from the children's hospital belonged mainly to B5/H24RxC, with only one isolate (18-10) being to B4/H24RxC. And they were all in BAP1 group and could not be further separated at level 2 using the global ML tree (reference genome 020026, CP034954 to CP034958). However, using the smaller phylogeny constructed only for the ST410 isolates from the children's hospital with isolate 19-7 (B5/H24RxC, Genbank: CP123017 to CP123023) as reference, fastbaps classified these isolates into three groups regardless of using filtered or unfiltered core SNPs alignment. Isolate 18-4 was classified into the same group as all other Group-2 isolates at level 1, but was separated from the group at level 2. The result supports the "Four groups" description in the manuscript for the children's hospital isolates. See Fig. S3c, 3d. Line 141-143.

In L142 of the 1st version of the manuscript, we in fact stated that "*All of the CREC ST410 isolates from the children's hospital outbreaks, including isolate 18-4, also belonged to group BAP1.*" The statement was only describing CREC ST410 isolates from the children's hospital while the red stars outside of BAP1 in Fig 3a represented isolates collected elsewhere in China in this study.

The reviewer was right to point out that some isolates other than B5/H24RxC isolates possessed *bla*_{NDM-5} gene (green ring in Fig 3a). But according to our analysis, both B4/H24RxC and B5/H24RxC possessed a F-type plasmid containing FII-1, FIA-1, and FIB-49 replicons (Fig3a, Fig S2). The plasmid consisted of a backbone and an antibiotic resistance region. The resistance region present in the B5/H24RxC clone included an IS26-flanked segment that contained *bla*_{NDM-5}. Although the *bla*_{NDM-5} gene was also present in some B4/H24RxC isolates, in those it was found in variants of the *bla*_{OXA-181} carrying X3 plasmid as previously reported [1, 2]. See Line 336-341.

1. Feng Y, Liu L, Lin J, et al. Key evolutionary events in the emergence of a globally disseminated, carbapenem resistant clone in the Escherichia coli ST410 lineage. *Commun Biol* 2019; 2: 322.
2. Chen L, Peirano G, Kreiswirth BN, Devinney R, Pitout JDD. Acquisition of genomic elements were pivotal for the success of Escherichia coli ST410. *J Antimicrob Chemoth* 2022.

- AMR screening. The authors screened for all genes in the ResFinder database but only reported a limited number of AMR genes. In some figures, genes mediating resistance to e.g. co-trimoxazole are named (Fig 5B) or total number of ARGs (Fig 3B) and in L100-101 eleven unspecified isolates are noted as carrying a *mcr-1* gene but the overall presence/ absence of AMR genes isn't reported. This could easily be a supp table/ supp figure and is useful background information for the reader. It would also help to address the point of MDR raised above in the intro. Further, in Fig6B, it appears as though *bla*_{CMY-2} (ESBL associated) is chromosomally integrated in the isolates shown. Is this gene present in all ST410? Could find no other mention of it in the manuscript or supplementary materials.

We thank the reviewer for their recommendation to use abritAMR in the comments in the method section. We have re-analysed the genomes of all Chinese CREC in this study using abritAMR and updated the method and result accordingly in the manuscript. We have also update Supplementary Data Set 1 to include all predicted ARGs and gene mutations that confer resistance for all Chinese CREC isolates.

We have also re-analysed the global collection of ST410 *E. coli* (n=956) for ARGs and gene mutations that confer resistance and presented the result in Supplementary Data Set 4. The result showed that 376/388 of the total number of B4/H24RxC and B5/H24RxC isolates possessed a copy of *bla*_{CMY-2} which was chromosomally integrated in the B4/H24RxC clone (ST410-B3 in [1]) as reported by Chen et al. [1]. The results suggested that *bla*_{CMY-2} was also chromosomally integrated in the B5/H24RxC clone. Although *bla*_{CMY-2} was not the focus of the

study, we showed it schematically in Fig 5b and S6. We apologise for not providing description to it in the figure legend which is now added.

1. Chen L, Peirano G, Kreiswirth BN, Devinney R, Pitout JDD. Acquisition of genomic elements were pivotal for the success of *Escherichia coli* ST410. *J Antimicrob Chemoth* 2022.

- The difference in virulence from the wax larvae model was shown between the B4 and B5 groups. Did the B5 isolates selected for testing have the HPI (reported in L156 as detected in 77.6% (135/174). Could there have been another reason apart from HPI for survival rate? e.g the authors note the difference in O-antigen and other differences in the accessory genome content.

B5/H24RxC isolates selected for the virulence assay have the HPI (Line 285-286), but we could not be certain if other factors are involved in the virulence without further investigation such as making isogenic deletion mutants to study the function of certain genes/factors. We regret to say that we are not able to further investigate this due to resource and time constraints, but we will be applying for funding to study these relevant questions in further research projects.

Discussion

- The authors call the new lineage 'hypervirulent' in the title but it was unclear as to why this lineage is hypervirulent/ what makes something hypervirulent vs virulent? The authors may wish to expand on this more as the authors state that the HPI element is frequently found in ExPEC and enables the bacteria to better colonise and persist (L302-306). It was unclear if the authors were also suggesting the HPI also mediated enhanced virulence? Or if the reason for hypervirulence was not yet determined?

In this study (Fig 7e), we found wax moth larvae infected with isolates of B5/H24RxC had a lower survival rate than those infected with one virulent control strain (hypervirulent *Acinetobacter baumannii* strain AB5075) and had a similar survival rate to those infected with the other virulent control strain (hypervirulent *Klebsiella pneumoniae* strain K1088), therefore we consider the new lineage 'hypervirulent'. Line 279-285.

It does appear that HPI mediates enhanced virulence in B5/H24RxC isolates, although we are not certain if there are other factors contributing to the virulence. We have now discussed this in the limitation section.

- The authors discuss differences in the presence of non-synonymous SNPs associated with B5/H24RxC (L317-325) and link through to some potential phenotypic changes. Unclear if this is accurate given not testing any of the SNPs explicitly. E.g. the significance of the biofilm formation and YeeJ discussion (L322-325) was unclear as the results (L272 -L275) stated both populations were poor biofilm performers and as such, the difference in YeeJ doesn't appear to be important.

We acknowledge that we had not validated any of the SNPs regarding their links to potential phenotypic changes. Although it is important to carry out such validation work, it does require significant effort and resources. Again, we regret to say that we are not able to further investigate this due to resource and time constraints. We have now discussed this in the limitation section.

As for the discussion regarding YeeJ, we have added a sentence to emphasise that its contribution to biofilm formation in these CREC clones remains unclear. Line 372-373.

- The authors suggest geographical spread from south-east Asia to other countries (L331-L333). While this is likely, it is a generalised statement that requires additional references (e.g from other studies demonstrating the emergence and dissemination of CREC / MDR *E. coli* in

the region) or additional work from the authors e.g. phylogeographical analyses/ ancestral state reconstruction to support this statement.

We agree that it is premature to suggest the geographical spread of B5/H24RxC clone from south-east Asia to other countries without additional work. Therefore, we have decided to remove the generalised statement from the manuscript as requested by the editor in their email accompanying the reviewers' reports.

- The strengths, limitations and future directions of this study were not really addressed in the discussion. A note was made of sampling bias (lack of data from companion animals and food samples) but the authors could consider these aspects of the study in greater detail.

We have included a section to discuss the strengths, limitation and future directions of this study. Line 390-399.

Figures + Tables

- There were 8 main figures included in the study + additional supplementary ones. Some of the main ones could be moved to supp e.g. Fig 4 – this was not a key finding to the study but more of a summary of the global dataset. Further, one of the key findings of the study was the novel O-antigen (novel1/OgN5) in the specific group. The comparison was shown in Supp Fig 2B but the authors could consider moving this panel to the main body of text in Fig 3 where they report the differences in O-antigens.

We have moved Fig 4 to supplementary (now Fig S1). We have also edited Fig 3 to include the comparison of the novel O-antigen Novel1 to a previously reported OgN5. Please see changes in Fig 3, Fig S1.

- For all violin plots – please also plot the underlying data

For all violin plots, we have now plotted the underlying data represented by the aligned dots in the figures.

- Fig 6B – the authors have a schematic showing what looks to be chromosomal integration of a blaCMY-2 gene. I could find no mention of the blaCMY-2 gene anywhere in the manuscript, supplementary figures or data tables.

We thank the reviewer for pointing this out. The blaCMY-2 gene was not found in all ST410 isolates; therefore, it is inaccurate to highlight it in the schematic representation of the chromosome. As this gene is not the focus of this study and it does not influence the categorisation of B4/H24RxC and B5/H24RxC clones, we have decided to keep it the schematic figures with a description added in the figure legend.

- Fig 2C : GrapeTree visualisation. What does the scale bar indicate? Why is 20_20 highlighted in red (this isolate is not referred to in the main manuscript but can see it was one of the isolates subject to ONT based off Supp Fig 3)?

In order to address reviewer 1's question 4 and to display the tree branches clearer, we now present the phylogeny in Figure 2c in a circular format instead of the figure generated using GrapeTree. The phylogeny is a maximum-likelihood core-genome SNP tree. We have also shown bootstrap values in the phylogeny represented by gradient colours on the branches.

The reason to highlight isolates 20_20 in red was because it was the only isolate in Group 1 from year 2020. This was described in the manuscript Line 134-136 "*Group-1 included 20 isolates mainly from between March 2018 and August 2019, with the exception of isolate 20-20 from June 2020, ...*" However, we have decided to remove the highlight since it may cause some confusion

Description for Figure 2c in the figure legend has been changed accordingly.

- Supp Fig 1 – why showing *fyuA*, *fstI* insertions? Refer to *fstI* in Intro (L73) and methods but not in results/ discussion, and *fyuA* as one of the genes on the HPI – using it as a marker for the HPI?

FstI insertion was used to define ST410-B2 and ST410-B3 as proposed by Chen et al., and was presented in Fig S2. *fyuA*, the indicative gene for the presence of the high pathogenicity island (HPI) is also presented in Fig S2.

- Supp table 5 sheet 3 – this looks like a duplication of the data on sheet 1?

We thank the reviewer for pointing this out.

Yes, sheet 3 indeed is a duplication of the data on sheet 1. We have now removed sheet 3.

General

- Number of times use acronyms without expanding either at first use or at all. E.g ARG first used at L151 and explained on L391. CFU, first used at L501, not explained at all.

We thank the reviewer for pointing this out. We have explained the acronyms in the manuscript at their first use.

REVIEWERS' COMMENTS

Reviewer #1 (Remarks to the Author):

The authors have done an excellent job of responding to reviewers' concerns. I have no additional suggestions.

Reviewer #2 (Remarks to the Author):

The manuscript "Global emergence of a new hypervirulent carbapenem-resistant *Escherichia coli* ST410 clone" is a revision of NCOMMS-23-20255. The authors have made a clear effort to address the reviewer comments and added in new analyses to address some of the raised concerns.

Introduction and ethics.

The introduction was greatly improved and much more accessible to a non-expert in ExPEC and the relevant ethics was easy to find in the revised manuscript.

Methods

The authors clarified many points on the methods and identified some were beyond the scope of this study. Few points that should be quick to address.

1). Enterobase dates for ST410 isolates – I think there is a typo in the dates at L312. Currently it reads as '14 June 2022 to 27 Oct 2023'. Perhaps the authors meant an earlier month or 2022?

2). In the rebuttal, the author provide the rationale for selection of the five B5/H24RxC isolates for ONT and also now include the isolates names in Table S1. This information should also be included in the methods to make it easy for readers to work out the selection process. Suggest something like "All five isolates, E22 + others, belong to B5/H24RxC clone, they were specifically selected based on their genomic differences (i.e. covering/representing the range of that diversity) as suggested by their positions in the B5/H24RxC phylogeny" could be included at L433 -435.

3) Selection of isolates for growth curves and competition assays. In the rebuttal letter the authors state the isolates were selected "based on their availability and their genetic characteristics" and provide

useful information as to the logic for isolate selection. I would suggest including some of the information in this part of the methods and naming the isolates, as was done for the virulence (L567) and iron source growth (L581)

4). Please include some text on data visualisation – specifically what tools were used. Expanded on in Figures and Tables section below.

Results and Discussion

5) I think the term ‘hypervirulence’ needs to be better explained as I think this is the first time it has been used for describing E. coli clones, and I think the term will be unfamiliar to those working on E coli. It is more established for other species, as the authors note with their use of Kp and Ab for control strains. Some of the explanation in the rebuttal letter could be included in the results around L284-285 so that it is very clear what this definition of hypervirulence is based upon data relative to Ab and Kp. I think it could be noted that this that this is the first time that this definition of ‘hypervirulence’ has been used for E. coli, so future studies would have a clear basis for potentially defining new clones as hypervirulent in the future.

6). I appreciate the authors including a paragraph on the limitations of this study. However, it would be useful to work the limitations into the appropriate sections of the discussion. E.g. Lines L392 -394 be included in the paragraph starting L347 so that some caution is provided in discussing the findings.

Figures and Tables

7). Can the authors include the relevant tools to generating the figures. E.g. the authors cite iTOL but unclear e.g. how the violin plots, map plots or genome comparisons were generated? Were these done in R with e.g. ggplot2, ggmap and gggenome? Supp Fig5 is from phandango but this isn't cited. Please ensure that the appropriate tools / packages used for data visualisation are also cited.

8). Violin plots – I can see dots are added but these don't appear to show the underlying data points but instead have a dot at each interval. E.g. Fig 1f.

REVIEWERS' COMMENTS

Reviewer #1 (Remarks to the Author):

The authors have done an excellent job of responding to reviewers' concerns. I have no additional suggestions.

Reviewer #2 (Remarks to the Author):

The manuscript “Global emergence of a new hypervirulent carbapenem-resistant *Escherichia coli* ST410 clone” is a revision of NCOMMS-23-20255. The authors have made a clear effort to address the reviewer comments and added in new analyses to address some of the raised concerns.

Introduction and ethics.

The introduction was greatly improved and much more accessible to a non-expert in ExPEC and the relevant ethics was easy to find in the revised manuscript.

Methods

The authors clarified many points on the methods and identified some were beyond the scope of this study. Few points that should be quick to address.

1). Enterobase dates for ST410 isolates – I think there is a typo in the dates at L312. Currently it reads as '14 Jan 2022 to 27 Oct 2023'. Perhaps the authors meant an earlier month or 2022?

We thank the reviewer for pointing out this typo. It should be '14 Jan 2022 to 27 Sept 2023'. We have made the correction in the manuscript.

2). In the rebuttal, the author provide the rationale for selection of the five B5/H24RxC isolates for ONT and also now include the isolates names in Table S1. This information should also be included in the methods to make it easy for readers to work out the selection process. Suggest something like “All five isolates, E22 + others, belong to B5/H24RxC clone, they were specifically selected based on their genomic differences (i.e. covering/representing the range of that diversity) as suggested by their positions in the B5/H24RxC phylogeny” could be included at L433 -435.

We have added the information in the methods as suggested. Please see Line 433-436.

3) Selection of isolates for growth curves and competition assays. In the rebuttal letter the authors state the isolates were selected “based on their availability and their genetic characteristics” and provide useful information as to the logic for isolate selection. I would suggest including some of the information in this part of the methods and naming the isolates, as was done for the virulence (L567) and iron source growth (L581)

We have added the information in the methods as suggested. Please see Line 537-539.

4). Please include some text on data visualisation – specifically what tools were used. Expanded on in Figures and Tables section below.

A 'Data visualisation' section has been added to the manuscript. Please see Line 631-636.

Results and Discussion

5) I think the term 'hypervirulence' needs to be better explained as I think this is the first time it has been used for describing *E. coli* clones, and I think the term will be unfamiliar to those working on *E. coli*. It is more established for other species, as the authors note with their use of Kp and Ab for control strains. Some of the explanation in the rebuttal letter could be included in the results around L284-285 so that it is very clear what this definition of hypervirulence is based upon data relative to Ab and Kp. I think it could be noted that this is the first time that this definition of 'hypervirulence' has been used for *E. coli*, so future studies would have a clear basis for potentially defining new clones as hypervirulent in the future.

We have added the following sentence in the manuscript to show that the definition of hypervirulence in B5/H24RxC is relative to hypervirulent *Klebsiella pneumoniae* strain K1088 and hypervirulent *Acinetobacter baumannii* strain AB5075. Also see Line 282-284.

"Based on this data relative to these two hypervirulent bacterial strains, we consider B5/H24RxC is a hypervirulent E. coli clone."

6). I appreciate the authors including a paragraph on the limitations of this study. However, it would be useful to work the limitations into the appropriate sections of the discussion. E.g. Lines L392 -394 be included in the paragraph starting L347 so that some caution is provided in discussing the findings.

The limitations of this study have been worked into the appropriate sections of the discussion. Please see Line 355-357, 373-375.

Figures and Tables

7). Can the authors include the relevant tools to generating the figures. E.g. the authors cite iTOL but unclear e.g. how the violin plots, map plots or genome comparisons were generated? Were these done in R with e.g. ggplot2, ggmap and gggenome? Supp Fig5 is from phandango but this isn't cited. Please ensure that the appropriate tools / packages used for data visualisation are also cited.

A data visualisation section has been added to the manuscript. Please see Line 631-636. Tools / packages used for data visualisation are now all cited.

8). Violin plots – I can see dots are added but these don't appear to show the underlying data points but instead have a dot at each interval. E.g. Fig 1f.

The violin plot Fig.1f is used to show the density of the MIC data of all test CREC isolates at different values. The widths of the plot indicate the frequency of an MIC reading is observed in all tested isolates. Multiple isolates (sometimes a large number) have the same MIC value. Showing all underlying data results in a solid line instead of separated dots (overlapping due to the restriction by the figure size), which will affect the readability of the lines indicating the quartiles and median. Therefore, only a dot (representing all dots with the same value) is show at each interval. As showing all underlying data doesn't add more meaningful information to the figure in this case, we have decided to keep the current violin plots (same explanation applied to Fig 3b,3c) without further modification.